# Hollow-core fibers with reduced surface roughness and ultralow loss in the short-wavelength range

Jonas H. Osório[1], Foued Amrani[1,2], Frédéric Delahaye[1,2], Ali Dhaybi[1], Kostiantyn Vasko [1], Federico Melli[3], Fabio Giovanardi[3], Damien Vandembroucq[4], Gilles Tessier[5], Luca Vincetti [3], Benoît Debord[1,2], Frédéric Gérôme[1,2] & Fetah Benabid [1,2] ✉

While optical fibers display excellent performances in the infrared, visible and ultraviolet ranges remain poorly addressed by them. Obtaining better fibers for the short-wavelength range has been restricted, in all fiber optics, by scattering processes. In hollow-core fibers, the scattering loss arises from the core roughness and represents the limiting factor for loss reduction regardless of the cladding confinement power. Here, we report on the reduction of the core surface roughness of hollow-core fibers by modifying their fabrication technique. The effect of the modified process has been quantified and the results showed a root-mean-square surface roughness reduction from 0.40 to 0.15 nm. The improvement in the core surface entailed fibers with ultralow loss at short wavelengths. The results reveal this approach as a promising path for the development of hollow-core fibers with loss that can potentially be orders of magnitude lower than the ones achievable with silica-core counterparts.

Hollow-core photonic crystal fibers (HCPCFs) approach their 30th anniversary[1]. Their remarkable performances, demonstrated in both fundamental and applied fields, justify the great efforts devoted by the HCPCF community toward a better understanding of their properties, the optimization of their designs and fabrication processes, as well as the consolidation of the application fields. Historically, photonic bandgap (PBG) fibers[1] were the first HCPCFs to appear as an alternative to overcome the fundamental constraint imposed by the Rayleigh scattering limit of glass and attain ultralow loss, particularly in the visible and ultraviolet spectral ranges, where silica attenuation dramatically increases. PBG fibers, however, have been dismissed as eligible candidates to surpass solid-core silica fibers' loss levels due to limitations such as the strong core-cladding optical overlap, the presence of surface modes, and their core surface roughness. In addition, having PBG fibers guiding at short wavelengths (<1 μm) requires smaller cladding pitches, which further complicates PBG

fibers' fabrication processes. The lowest loss reported for PBG fibers is 1.2 dB/km around 1600 nm[2]. At shorter wavelengths, the minimum attenuation is much higher: 870 dB/km at 557 nm[3].

Alternatively, inhibited-coupling (IC) guiding fibers[4] have been proven to exhibit confinement loss (CL) figures comparable to PBG fibers but with the outstanding difference that the core-cladding optical overlap is several orders of magnitude smaller. Moreover, as IC fibers work on the large pitch regime, greater cladding pitches can be used to operate at short wavelengths. Furthermore, the criteria for IC guidance cause the fiber CL to be strongly dependent on the core contour shape. It inspired, for example, the proposal of the hypocy-cloid core contour (negative curvature) in 2010[5,6], which provided a dramatic reduction of the loss in IC fibers. Also, the demonstration of guidance in single-ring tubular lattice (SR-TL) HCPCFs[7] stands out as an important achievement that motivated further developments in HCPCF technology.

[1]GPPMM Group, XLIM Institute, CNRS UMR 7252, University of Limoges, Limoges 87060, France. [2]GLOphotonics, 123 Avenue Albert Thomas, Limoges 87060, France. [3]Departament of Engineering "Enzo Ferrari", University of Modena and Reggio Emilia, Modena 41125, Italy. [4]PMMH, CNRS UMR 7636, ESPCI Paris, PSL University, Sorbonne University, Paris Cité University, Paris 75005, France. [5]Vision Institute, CNRS UMR 7210, Sorbonne University, Paris 75012, France. ✉e-mail: f.benabid@xlim.fr

The current state-of-the-art loss levels in IC fibers are set by the surface-roughness scattering loss (SSL) for fibers guiding at short wavelengths (<1 μm) and by the fiber design for fibers guiding at longer ones[8,9]. Concerning the latter, alternative cladding design to kagome and SR-TL HCPCF has entailed ultralow loss figures in the infrared range[10–13]. On the other hand, for shorter wavelengths, SR-TL HCPCFs have provided loss values of 7.7 dB/km at 750 nm and 13.8 dB/km at 539 nm[7,14]. Nested-tubes HCPCFs, in turn, display loss figures of 0.6 dB/km at 850 nm and 2.85 dB/km at 660 nm[12,15]. Finally, conjoined-tubes HCPCFs show the figures of 3.8 dB/km at 680 nm and 4.9 dB/km at 558 nm[16]. In the UV range, loss figures values of 100 dB/km at 218 nm and 130 dB/km at 300 nm have been measured[17,18].

In this framework, although improvements in the cladding design of IC guiding fibers have allowed decreasing the loss figures in the infrared range, obtaining fibers for the visible and ultraviolet range remains a more challenging task due to the SSL limitation. Indeed, SSL is set by the core surface roughness, which arises from thermal surface capillary waves (SCW) that are frozen during the fiber draw[2]. SSL assumes the form of $\alpha_{SSL} = \eta \times F \times (\lambda/\lambda_0)^{-3}$ (where $F$ is the core mode optical overlap with the core contour, $\lambda$ is the wavelength, and $\lambda_0$ is a calibrating constant) and scales quadratically with the surface roughness root-mean-square (rms) height, i.e., $\alpha_{SSL} \propto h_{rms}^2$. Indeed, the factor $\eta$ in the SSL formula relates to the surface quality, being proportional to $h_{rms}^2$. Reducing the SSL, therefore, implies controlling and/or reducing the roughness height. In this context, despite recent results showing that increasing the drawing stress during the fabrication process of silica capillaries diminishes the roughness along the drawing direction[19], no work has been reported so far on HCPCFs and on how to mitigate SSL-dominated scenarios.

Here, we report that, by revisiting the HCPCFs fabrication technique, reduction of the rms roughness of the core surfaces of HCPCFs is possible. Our innovative technique incorporates the concept that shear flow can attenuate capillary waves[20,21] in the fiber drawing process. Having counter-directional gas and glass fluxes during the fiber fabrication allowed to attain increased shear rate on the membranes of SR-TL HCPCFs while preserving the fiber structural integrity and suitable microstructure geometrical dimensions. In our investigation, we studied two sets of fibers, the first one produced by using the standard HCPCF fabrication method and the second one by using this innovative technique. Optical profilometry measurements showed that the rms roughness was reduced from 0.40 nm down to 0.15 nm via the utilization of the technique proposed herein. The reduction of the core surface roughness allowed to obtain fibers with loss figures as low as 50 dB/km at 290 nm, 9.7 dB/km at 369 nm, 5.0 dB/km at 480 nm, 0.9 dB/km at 558 nm, and 1.8 dB/km at 719 nm. We believe that our results provide a new framework for the development of optical fibers transmitting light in the visible and ultraviolet ranges, and open exciting prospects in UV-photonics.

## Results

### Surface capillary waves within HCPCFs' scenario

During HCPCF fabrication, the structured glass preforms undergo heating inside a high-temperature furnace. The heating process entails melting of the preform, which allows, by suitably pulling the fiber and pressurizing its internal microstructure, to successfully fabricate the desired HCPCF architecture. Within the heating process context, the dynamics of SCW is established as a result of two competing effects, namely the thermal noise, which is prone to ruffle the surface, and the interface tension, which tends to attenuate the SCW oscillations. Indeed, the SCW dynamics has been proven to govern the surface roughness of the glass surfaces in HCPCFs, as they are abruptly frozen at the glass transition temperature, $T_G$[2]. Under the SCW framework, the Fourier spectrum of the surface height profiles, which is typically assessed via their power spectral density (PSD) functions calculated over 1D surface profiles, is expected to display a $1/f$ behavior, where $f$ is

the spatial frequency. The $1/f$-trended behavior of PSD functions accounted from HCPCFs core surface profiles has been observed in previous works[2,21–23]. For low spatial frequency values (<$5 \times 10^{-2}$ μm$^{-1}$), however, deviation from the $1/f$ trend has been observed, and PSD functions following a $1/f^3$-trend have been detected[23]. The latter is likely to be due to boundary conditions imposed by the heat zone length of the drawing furnace.

As mentioned before, we here develop a technique for reducing the roughness of HCPCFs core surfaces to attain ultralow loss figures in the short-wavelength range. In our study, we employ eight-cladding tubes SR-TL HCPCFs due to their potential to provide ultralow loss in the visible and ultraviolet spectral ranges allied to a simple fiber structure from a fabrication viewpoint. Thus, hereafter all the fibers studied in the present work are eight-cladding tubes SR-TL HCPCFs. Figure 1a presents the cross-section of one of the SR-TL HCPCFs used in our investigations, which has been fabricated by using standard HCPCF fabrication methods. To assess the quality of the core surfaces in the fiber, optical profilometry and atomic force microscopy (AFM) have been employed. Figure 1b shows typical height profiles ($\Delta h$) along the fiber axis measured by using AFM (top plot) and a picometer-sensitivity optical profilometer (bottom plot)[22,23]. Figure 1c, in turn, presents a representative AFM-measured surface profile taken for a 6 μm × 6 μm area on the fiber core surface. The optical profilometer and AFM data allow obtaining the PSD plot shown in Fig. 1d. Optical profilometry is limited to low spatial frequencies (a few $10^{-1}$ μm$^{-1}$) by diffraction and probe spot spacing. Conversely, the scanning ranges limit AFM to higher frequencies, between 1 and a few tens of μm$^{-1}$. The combination of the two measurement techniques provides a spatial frequency interval ranging from $1 \times 10^{-2}$ μm$^{-1}$ to 20 μm$^{-1}$. For comparison, we include in the plot a $1/f$ trend calculated from Eq. (1) given below[2]:

$$|H_{SCW}(f)|^2 = \frac{k_B T_G}{2\pi\gamma f} \tag{1}$$

where $k_B$ is the Boltzmann constant and $\gamma$ is the surface tension, which stands for the PSD expected from 1D surface profiles under the SCW scenario. Here, we used $T_G/\gamma = 2000$ K.m$^2$/J, which is consistent with the $T_G$ and $\gamma$ values reported in ref. [2] for drawn silica capillaries. It is seen that the PSD trend deviates from the $1/f$ behavior for $f < 1 \times 10^{-2}$ μm$^{-1}$, similar to what has been found in recent investigations[22,23]. While the $1/f$ behavior directly results from the freezing of SCW, the deviation of it at lower frequencies can be ascribed to boundary size effects. As an additional comparison, we show in Fig. 1d a modified trend (green dashed line in Fig. 1d, attained when quantizing the SCW transversally), obtained when multiplying Eq. (1) by $\coth\left(\frac{Wf}{2}\right)$, where $W$ is the perimeter of the cladding tubes[2]. The latter is calculated here for cladding tubes with 15 μm diameter, a typical value in our fibers. While the trend modulated by the hyperbolic cotangent term allows calculating larger PSD values at low spatial frequencies compared to the $1/f$ trend, the measured PSD remains larger than it, consistently with what has been reported previously[23].

### Shear stress as a means to structure the surface roughness profile

The inherent roughness of glass surfaces arises from fluctuations of capillary waves, which, as discussed in the last section, result from the interplay between the thermal noise and the glass interface tension[24]. In the absence of shear, the rms height of these frozen fluctuations amounts to $\sqrt{\left(\frac{k_B T_G}{2\pi\gamma}\right)\ln\left(\frac{\Lambda_M}{\Lambda_m}\right)} \approx 0.4$ nm, which is obtained when using $T_G = 1500$ K, $\gamma \approx 0.3$ J/m$^2$, and the upper and lower spatial cutoffs, $\Lambda_M$ and $\Lambda_m$, associated respectively with the silica capillary length (4 mm) and molecular length (0.5 nm)[25]. This rms height level is, hence, referenced as thermodynamic equilibrium surface roughness.

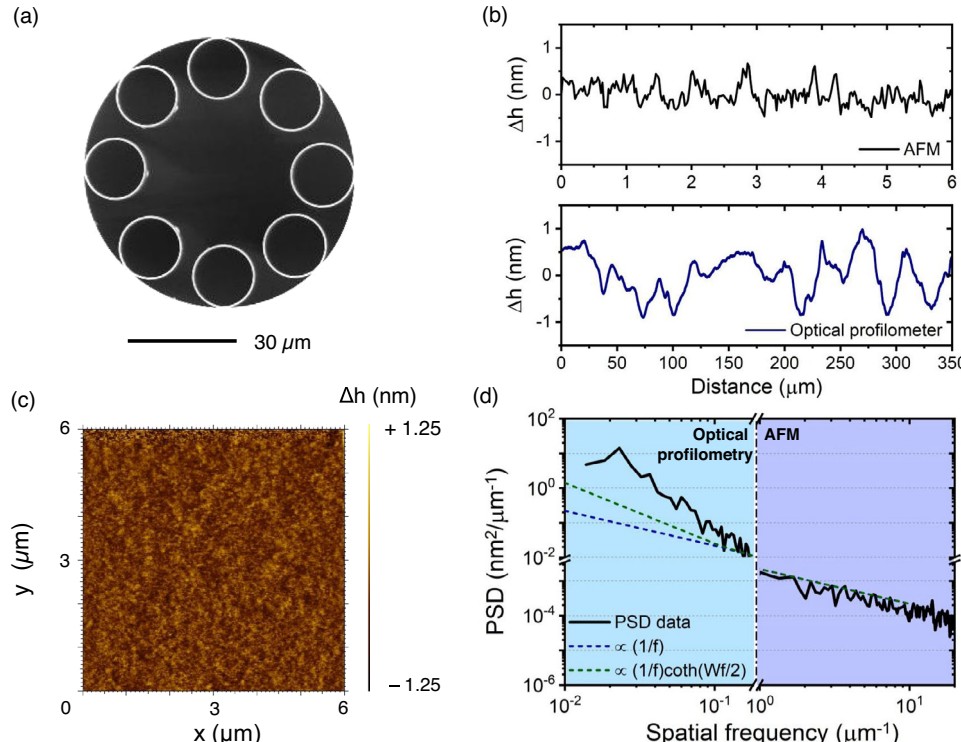

**Fig. 1 | HCPCF core surface characterization. a** Cross-section of an SR-TL HCPCF. **b** Typical surface profiles for a fiber drawn by using typical HCPCF fabrication methods measured by AFM (top plot) and optical profilometry (bottom plot). **c** Typical AFM measurement result for a fiber drawn by using typical HCPCF fabrication methods. **d** PSD plot accounted from optical profilometry and AFM measurements and PSD trend lines. AFM atomic force microscopy, PSD power spectral density.

In 2006, Derks et al.[21] concluded that shear can suppress capillary waves and make interfaces smoother. To describe this effect, the authors introduced the concept of effective interfacial tension, $\gamma_{eff}$, to be assigned to sheared systems, which increases with the shear rate. According to the model proposed in ref. [21], the effective interfacial tension can be expressed by Eq. (2), where $\gamma_0$ is the interfacial tension at zero shear and $\Phi(\kappa)$ is a positive function that grows with increasing shear rate $\kappa$.

$$\gamma_{eff}(\kappa) = \gamma_0 + \Phi(\kappa) \qquad (2)$$

If one considers a glass surface surrounded by air, the function $\Phi(\kappa)$ will be given by Eq. (3), where $\mu_{glass}$ and $\mu_{air}$ are the viscosities of glass and air, respectively, and $L_{cap}$ is the capillary length, given by $\sqrt{\gamma_0 / g\left(\rho_{glass} - \rho_{air}\right)}$, where $g$ is the acceleration of gravity, and $\rho_{air}$ and $\rho_{glass}$ are the densities of air and glass, respectively[21].

$$\Phi(\kappa) = \left(\frac{3k_B T}{4\pi}\right)\left[\frac{\left(\mu_{air} + \mu_{glass}\right)\kappa}{\gamma_0 L_{cap}}\right]\sqrt{\left[\frac{\left(\mu_{air} + \mu_{glass}\right)\kappa L_{cap}}{\gamma_0}\right]^2 - 1} \quad (3)$$

As the mean square roughness, $\langle h^2 \rangle$, is inversely proportional to the interfacial tension, Derks et al. stated that the amplitude of the capillary waves can be expressed as Eq. (4), where $\langle h^2 \rangle(\kappa = 0)$ is the mean square roughness under zero shear[21].

$$\langle h^2 \rangle(\kappa) = \left[\frac{\gamma_0}{\gamma_{eff}(\kappa)}\right]\langle h^2 \rangle(\kappa = 0) \qquad (4)$$

The expression displayed above shows that the mean squared height of sheared surfaces is expected to be decreased relative to the value corresponding to not sheared ones by a factor $\gamma_{eff}/\gamma_0$.

Such a reduction of interfacial fluctuations has also been studied by Thiébaud et al.[20] and Smith et al.[26,27]. Particularly, Smith et al.[26,27] performed simulations on the dynamics of laterally driven surfaces and concluded that the existence of shear implies the reduction of the interfacial width. They affirmed that shear acts as an effective confinement force in the system which can suppress the interfacial capillary wave fluctuations. Similar to ref. [21], they argued that defining a "none-equilibrium surface tension" to theoretically fit the behavior of the height-height correlation functions of the sheared surfaces entails an increase of such tension as the system is more strongly driven.

Within the context of optical fibers, the reduction of the roughness of sheared glass surfaces has been assessed by Bresson et al. on glass tubes[19]. In the latter, by evaluating the surface profiles of the glass tubes with typical diameters of 220 µm and thicknesses of 15 µm, the authors identified that the surface roughness levels can be lowered by the fiber drawing process thanks to the attenuation of the SCW along the drawing direction. In this context, they detected an anisotropic behavior of the height correlations in the fabricated fibers and showed that the glass surfaces can retain a structural signature of the direction of the flow that took place during the fiber fabrication.

The parameter chosen in ref. [19] for studying the SCW attenuation, the tension experienced by the fiber during fabrication, results however from several draw parameters such as furnace temperature, draw speeds, the dimensions of fiber to be drawn and of its preform. Hence, although it had allowed the authors in ref. [19] to correlate the lessening of the surface roughness levels in the studied glass tubes with the drawing process, such a parameter does not present itself as the most adequate indicator of flow establishment inside HCPCFs for our

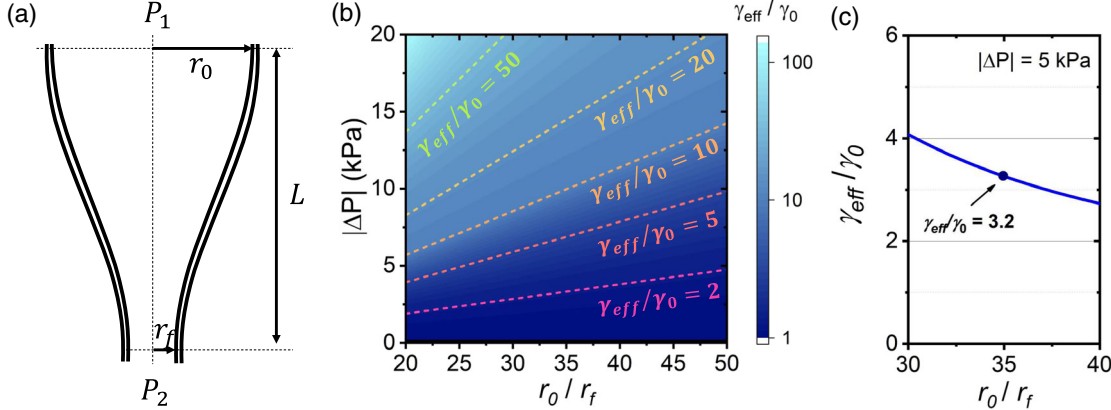

**Fig. 2 | Toy model for the shear rate on the fiber core surfaces. a** Diagram of the transition region between the preform and the fiber. $P_1$ and $P_2$ stand for the pressure in the preform and in the fiber, respectively; $r_0$ and $r_f$ are the preform and fiber radii, respectively. $L$ is the preform-to-fiber transition length. **b** Color plot of $\frac{\gamma_{eff}}{\gamma_0}$ for varying pressure differences, $|\Delta P|$, and preform-to-fiber down-ratio, $\frac{r_0}{r_f}$. **c** Graph for $\frac{\gamma_{eff}}{\gamma_0}$ as a function of $\frac{r_0}{r_f}$ for $|\Delta P| = 5$kPa.

investigation, whose aim is to study the reduction of the surface roughness levels inside SR-TL HCPCFs.

Thus, in order to establish a flow inside the microstructure of SR-TL HCPCFs, we have worked with the application of pressure gradients inside the preforms during fiber fabrication, as described in the following. To study the motion of the gas inside the fiber during its fabrication and, thus, provide a toy model to account for the shear rate levels inside the fiber core, one can approach Navier–Stokes and continuity equations applied to the flow of a gas inside a tube with varying diameters[28]. If one hypothesizes on having a fully developed laminar flow scenario and assumes that the transition region between the preform and the fiber has an exponential shape, the velocity of the gas along the axis of the fiber ($z$-direction), $u_z$, can be written as in Eq. (5), where $\Delta P = P_1 - P_2$ is the pressure variation between the preform and fiber regions ($P_1$ and $P_2$, respectively), $L$ is the preform-to-fiber transition length, and $r$ is the radial coordinate. The parameter $\xi$, in turn, considers the exponential shape of the preform-to-fiber transition and is given by Eq. (6), where $r_0$ and $r_f$ are the preform and fiber radii, respectively[29,30]. Figure 2a provides a diagram of the transition region between the preform and the fiber.

$$u_z = \frac{|\Delta P|}{4\mu_{air}L}\left[r_0^2\exp\left[-2\xi\left(\frac{z}{L}\right)\right] - r^2\right] \quad (5)$$

$$\xi = \ln\left(\frac{r_0}{r_f}\right) \quad (6)$$

We remark that the laminar flow hypothesis is sound as the calculated Reynolds number for the gas flow we study herein has an order of magnitude of $10^{-4}$. In addition, we mention that our toy model considers no glass flow. This is also a sound hypothesis as the axial velocity of the glass is considerably lower than the speed of the gas inside the fiber (the ratio between the axial velocities of the gas and the glass flows is estimated to be ~30). Finally, in our model, we consider no collapse or expansion of the glass. This assumption is valid because we control the pressure levels inside the fiber to avoid glass collapsing, and, although in our draws we use pressure to inflate the microstructure holes, the rate of such expansion is small. These considerations allow us to use the negative exponential to describe the shape of the preform-to-fiber transition region.

Equation (5) allows calculating the volumetric flow rate at a certain position $z$ and, subsequently, the shear rate κ associated with such a

flow. The resulting expression is shown in Eq. (7).

$$\kappa = \frac{|\Delta P| r_0\exp\left(-\frac{\xi z}{L}\right)}{2\mu_{air}L} \quad (7)$$

This, in turn, allows estimating the effective interfacial tension of the sheared surface of the tube wall by using Eq. (2). Figure 2b shows a color plot of $\frac{\gamma_{eff}}{\gamma_0}$ for typical ranges of pressure gradient, $|\Delta P|$, and fiber and preform radius ratios, $\frac{r_0}{r_f}$, during HCPCF drawing. By observing Fig. 2b, one sees that the toy model one approaches herein provides a working range for the fiber drawing parameters entailing increased $\frac{\gamma_{eff}}{\gamma_0}$ ratio (e.g., pressure values, preform, and fiber dimensions), which are expected to impact $\langle h^2 \rangle$ values. In particular, in Fig. 2c, we present a graph of $\frac{\gamma_{eff}}{\gamma_0}$ as a function of $\frac{r_0}{r_f}$ for $|\Delta P| = 5kPa$, where we can observe that for $\frac{r_0}{r_f} = 35$, a typical down-ratio value in our fiber drawings, $\frac{\gamma_{eff}}{\gamma_0} = 3.2$, which stands for the expected reduction ratio on $\langle h^2 \rangle$ values, according to Eq. (4). In the graph shown in Fig. 2c, we used $T = 2300K$, $L = 2$cm, $\mu_{air} = 5.817 \times 10^{-5}$Pa · s, $\mu_{glass} = 6.165 \times 10^3$Pa · s, $\gamma_0 = 0.3$N/m, $\rho_{air} = 0.199$kg/m³, and $\rho_{glass} = 2200$kg/m³.

While the above model is satisfactory in demonstrating the establishment of a shear flow, an essential condition for surface roughness rms reduction, and gives a qualitative $\langle h^2 \rangle$ reduction factor consistent with the experimental results to be described in the next sections, we signal that such a simplified model certainly does not consider all the complexity of the fiber drawing. Furthermore, the choice of the parameters in making the preform and the fibers must ensure both the fiber structural integrity and the establishment of the necessary shearing during the fiber draw. Consequently, the fiber design and drawing follow a highly demanding trade-off between the different preform and fiber drawing parameters, an experimental work that has been very carefully designed and carried out by us. In our investigation, and considering the limited range of the fiber drawing parameters, the design of fiber fabrication entails adequate fiber microstructures, preform and fiber dimensions, fiber drawing speeds, as well as the pressure values to be applied in the internal microstructure of the fibers during fabrication, able to accommodate the shear flow conditions and attain fibers with optimized performances.

**Surface roughness reduction in SR-TL HCPCFs**
The usual technique for drawing SR-TL HCPCFs employs the pressurization of core and cladding regions, so adequate dimensions are achieved on the fiber cross-section. Current techniques, thus, uses co-directional gas and glass flows during fabrication. Differently,

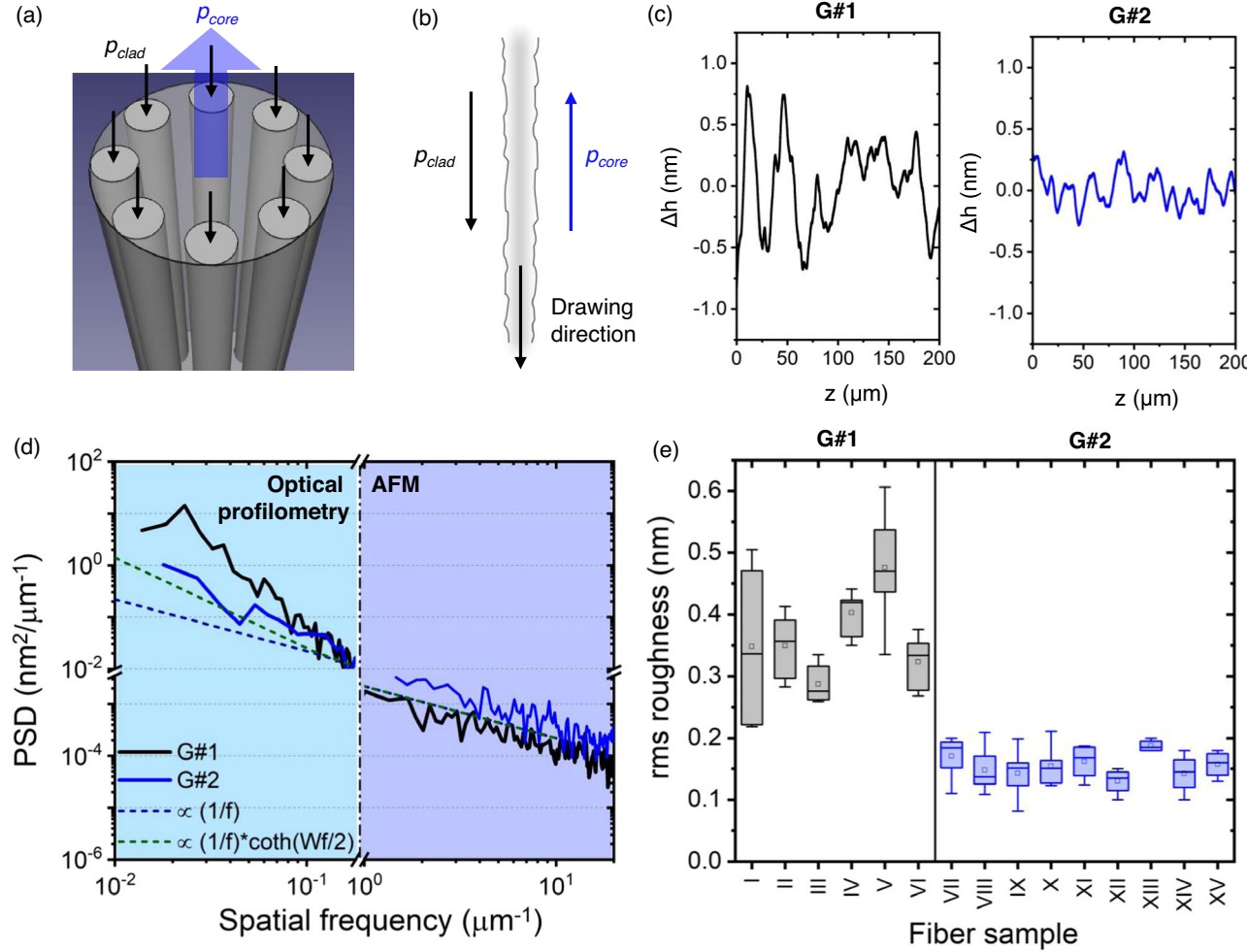

**Fig. 3 | Core surface roughness reduction in HCPCFs. a** Diagram for the pressure application in the fiber preform during the innovative process proposed in this manuscript; $p_{core}$ pressure in the fiber core, $p_{clad}$ pressure in the cladding tubes. **b** Diagram of the interfaces between the surfaces of the cladding tubes, the tubes internal region and the fiber core. **c** Typical roughness profiles along the fiber axis and **d** PSD for fibers in G#1 and G#2. **e** Rms roughness for fibers in G#1 and G#2 (sample numbering according to Table 1). AFM atomic force microscopy, PSD power spectral density.

here we propose the utilization of counter-directional gas and glass flows within the fiber microstructure during the fiber draw for adding shear to the microstructure's glass membranes and, hence, attaining smoother core surfaces. In our technique, a vacuum is applied to the fiber core, while the ring of tubes in the SR-TL HCPCF structure is pressurized so to achieve adequate fiber geometrical parameters (Fig. 3a). Figure 3b presents a diagram for the interfaces between the surfaces of the cladding tubes, the fiber core, and the internal region of the cladding tubes. During the fabrication routines following our methods, the fiber outer diameter variation was no larger than 0.5%. Also, evaluation of the fiber microstructure elements allowed for ascertaining a highly consistent fiber cross-section along the 500–1000 m typically fabricated lengths.

To demonstrate the reduction of the core surface roughness via the utilization of the approach proposed herein, we systematically studied two sets of SR-TL HCPCFs, both fabricated by using the stack-and-draw technique. The first group (G#1, composed of five different SR-TL HCPCFs fabricated in independent fiber draws, with cladding tubes thickness ranging from 230 to 580 nm) was produced by using the standard fabrication method for HCPCFs. Fibers in the second group (G#2, composed of nine different SR-TL HCPCFs fabricated in independent fiber draws, with cladding tubes thickness ranging from 300 to 1220 nm), instead, were produced by the innovative technique reported herein, i.e., during fiber fabrication, controlled vacuum

was applied to the fiber core while pressure was used to inflate the cladding tubes. Table 1 presents the geometrical dimensions of the fibers in G#1 and G#2. It is worth mentioning that to adequately perform our analyses, we have endeavored to have, in G#1 and G#2, fibers with comparable cladding tube dimensions, as one can verify in Table 1.

The core roughness profiles along the propagation axis of the fibers in G#1 and G#2 have been experimentally measured by using a picometer-resolution optical profilometer[22,23]. Figure 3c shows typical surface profiles for fibers in G#1 (left-hand side) and G#2 (right-hand side) measured in the optical profilometry experiments. It is seen that fibers in G#2 present reduced peak-to-peak roughness values compared with fibers in G#1 (for fibers in G#1, peak-to-peak values are around 1.5 nm, and, for fibers in G#2, around 0.5 nm). Moreover, Fig. 3d exposes typical PSD traces for fibers in G#1 and G#2. Noteworthily, the fabrication methods reported herein entail a reduction of the PSD values at spatial frequencies lower than $10^{-1}\,\mu m^{-1}$, which, in turn, readily impacts the rms roughness of the fibers. It is worth mentioning that, although one observes larger PSD values at spectral frequencies greater than $1\,\mu m^{-1}$ for fibers in G#2 than for fibers in G#1, the PSD figures at larger spatial frequencies are orders of magnitude lower than the corresponding PSD values at lower spatial frequencies. It entails that the spectral components of the roughness at higher frequencies have a significantly reduced impact on the rms surface

**Table 1 | G#1 and G#2 fibers' parameters**

| G#1 | | | | G#2 | | | |
|---|---|---|---|---|---|---|---|
| Fiber sample | $t$ (nm) | $D_{tubes}$ (µm) | $D_{core}$ (µm) | Fiber sample | $t$ (nm) | $D_{tubes}$ (µm) | $D_{core}$ (µm) |
| I | 230 | 15.6 | 41.5 | VII | 300 | 10.6 | 28.0 |
| II | 415 | 13.0 | 30.0 | VIII | 305 | 10.7 | 29.5 |
| III | 465 | 11.2 | 32.0 | IX | 360 | 8.5 | 25.0 |
| IV | 500 | 10.8 | 34.5 | X | 600 | 11.2 | 27.0 |
| V | 545 | 16.0 | 39.9 | XI | 720 | 15.6 | 35.0 |
| VI | 580 | 10.2 | 37.0 | XII | 1000 | 16.5 | 40.0 |
| | | | | XIII | 1050 | 14.7 | 41.0 |
| | | | | XIV | 1080 | 14.1 | 40.0 |
| | | | | XV | 1120 | 14.0 | 42.0 |

The geometrical dimensions of the fibers in G#1 and G#2.
$t$ tubes' thickness, $D_{tubes}$ diameter of the cladding tubes, $D_{core}$ diameter of the core.

roughness compared with the components at lower spatial frequencies, and also on the SSL.

In this context, Fig. 3e shows box charts for the rms roughness values, accounted from the optical profilometry measurements, for fibers in G#1 (left-hand side) and G#2 (right-hand side). Values in Fig. 3e were obtained from the rms roughness values measured in several scans (with typical lengths of 200 µm) for the fiber samples in G#1 and G#2. It is seen that fibers in G#1 have rms roughness values that vary around 0.40 nm while fibers in G#2 have rms roughness values around 0.15 nm. Indeed, the utilization of our techniques allowed improving the quality of the core surface of the HCPCFs by a factor of 2.7 and attaining fibers with core surface roughness lower than the thermodynamic equilibrium level.

## Ultralow loss in SR-TL HCPCF in the short-wavelength range

As described in the last section, the utilization of our methods has entailed fibers with smoother core surfaces. A direct effect of the reduction of the roughness levels of the core surfaces is the lowering of the SSL in the fabricated fibers. In this framework, we report on SR-TL HCPCFs with ultralow loss in the short-wavelength range (i.e., $\lambda < 1$ µm).

Figure 4 summarizes the transmission loss of both groups of fibers along with the numerically simulated main sources of power attenuation during the optical propagation.

Figure 4a presents the loss spectra of representative fibers in G#1 and G#2 with similar cladding tube thicknesses ($t \sim 0.6$ µm). By observing the loss spectrum of G#1 fiber (black curve; core diameter 41 µm), one sees that there is a considerable loss increase for wavelengths lower than 600 nm. Otherwise, the results for G#2 fiber (blue curve; core diameter 27 µm) show that a decreasing loss trend is maintained even for wavelengths lower than 600 nm, and this is despite having a smaller core diameter and its subsequent higher CL. The difference between the loss values for the band between 600 and 1000 nm is likely due to the different core diameters of the two considered fibers. To further substantiate our observations, we plot, in Fig. 4b, the loss values of the representative fibers in G#1 and G#2 multiplied by $R_{co}^4$, where $R_{co}$ is the core radius. Multiplication by such a factor allows to normalize the fiber loss trends with respect to the core radii of the fibers[31] and provides a clearer visualization of the difference between the loss trends for fibers in G#1 and G#2. The distinction between the loss behaviors of fibers in G#1 and G#2 shown in Fig. 4b is, therefore, a strong indicator of the SSL reduction in G#2 fibers as one will further analyze in the following when considering the different loss mechanisms in HCPCFs.

In order to quantify the impact of the surface roughness considering the different transmission loss mechanisms, which are the CL,

SSL, bending loss, and microbending loss (MBL)[32], we have started by estimating the surface roughness impact on the CL. This was done by considering a transverse roughness-free SR-TL-HCPCF with a core radius of 13.5 µm and thickness of 0.6 µm (purple dashed line in Fig. 4c and labeled ideal structure) and comparing it with the same fiber but displaying a transverse roughness rms value of 0.15 nm (blue solid line in Fig. 4c and labeled t-roughness)−we clarify that, although we did not measure the roughness on the transverse direction, we assume that the transverse surface height variations follow the characteristics of the roughness measured along the fiber axis as its formation stems from a stochastic process. Observation of Fig. 4c allows verifying that the transverse roughness levels minorly impact the CL values in our operating conditions. We can, thus, disregard this influence in our following analyses.

The breakdown of the contributions of the different loss mechanisms accounted from the simulations (CL, SSL, MBL, and TL = CL + SSL + MBL) is presented in Fig. 4d, e (dashed and dotted curves), together with the measured loss spectra (solid curves). In the literature, the MBL is described according to $MBL = \beta_0^2 C(\Delta\beta_{01}) \left( \langle 0|x^2|0\rangle - |\langle 1|x^2|0\rangle|^2 \right)$, where $\beta_0$ is the propagation constant of the core fundamental mode, $C(\Delta\beta_{01}) = C_0/\Delta\beta_{01}^2$ is the PSD of the microbending stochastic process computed at $\Delta\beta_{01}$, the difference between the propagation constants of the core fundamental and LP$_{11}$-like modes, and $C_0 = 1$[32]. In turn, $\langle 0|x^2|0\rangle$ is the power lost by the core fundamental mode and $|\langle 1|x^2|0\rangle|^2$ is the power transferred from the fundamental mode to the LP$_{11}$-like modes[32]. Herein, we used $|\langle 1|x^2|0\rangle|^2 = 0$[33]. It is worth mentioning, however, that although we present the MBL contributions corresponding to our fibers so as to situate our results among the other studies available in the literature, we consider the description of the MBL in HCPCFs as a model still in construction which calls out for further studies by the HCPCF community. In addition, we mention that the CL values shown in Fig. 4d, e consider the bending loss corresponding to a curvature radius of 1 m, which have been the radius of the fiber coils in the loss measurements.

The most striking information one can assimilate from Fig. 4d, e concerns the SSL. Data in Fig. 4d (G#1 representative fiber) shows that the factor $\eta$ in the SSL expression ($\alpha_{SSL} = \eta \times F \times (\lambda/\lambda_0)^{-3}$, as described in the introduction) that entails adequate fitting between the simulated TL and the measured loss is $\eta_{G\#1} = 8.0 \times 10^{-3}$. In turn, the value of $\eta$ that allows attaining suitable fitting between the simulated TL and the measured loss for the representative fiber in G#2 is $\eta_{G\#2} = 1.1 \times 10^{-3}$. If one reminds that, as discussed in the introduction, $\eta$ can be related to the square of the rms surface roughness, the ratio between the square roots of the obtained $\eta$ can be readily associated with the reduction of the rms core surface roughness values in our fibers. Remarkably, $\sqrt{\frac{\eta_{G\#1}}{\eta_{G\#2}}} \approx 2.7$, which is consistent with our surface roughness measurement results. In the above-mentioned analyses, we used $\lambda_0 = 1700$ nm. Also, we mention that different $C_0$ values in the MBL formula have been considered (from 0 to 2.86[33]), and the ratio of the rms roughness suppression was found to be in the range of 2.3−2.7.

Finally, we report on record attenuation values for fibers guiding at short wavelengths. Figure 5a, b exhibits the cross-sections of the fibers we report here (referenced as Fiber A and Fiber B, both fabricated by following the innovative fabrication methods reported herein) and their corresponding measured loss. The minimum loss figures of Fiber A are 50.0 dB/km at 290 nm, 9.7 dB/km at 369 nm, and 5.0 dB/km at 480 nm. In turn, the minimum attenuation values for Fiber B are 0.9 dB/km at 558 nm and 1.8 dB/km at 719 nm. Table 2 displays the fibers' geometrical parameters.

Figure 5c contextualizes the results of Fiber A and Fiber B within the IC HCPCF framework in the wavelength interval between 250 and 900 nm. In addition, Fig. 5c presents the silica Rayleigh scattering limit trend (SRSL), which stands as a benchmark for situating our results

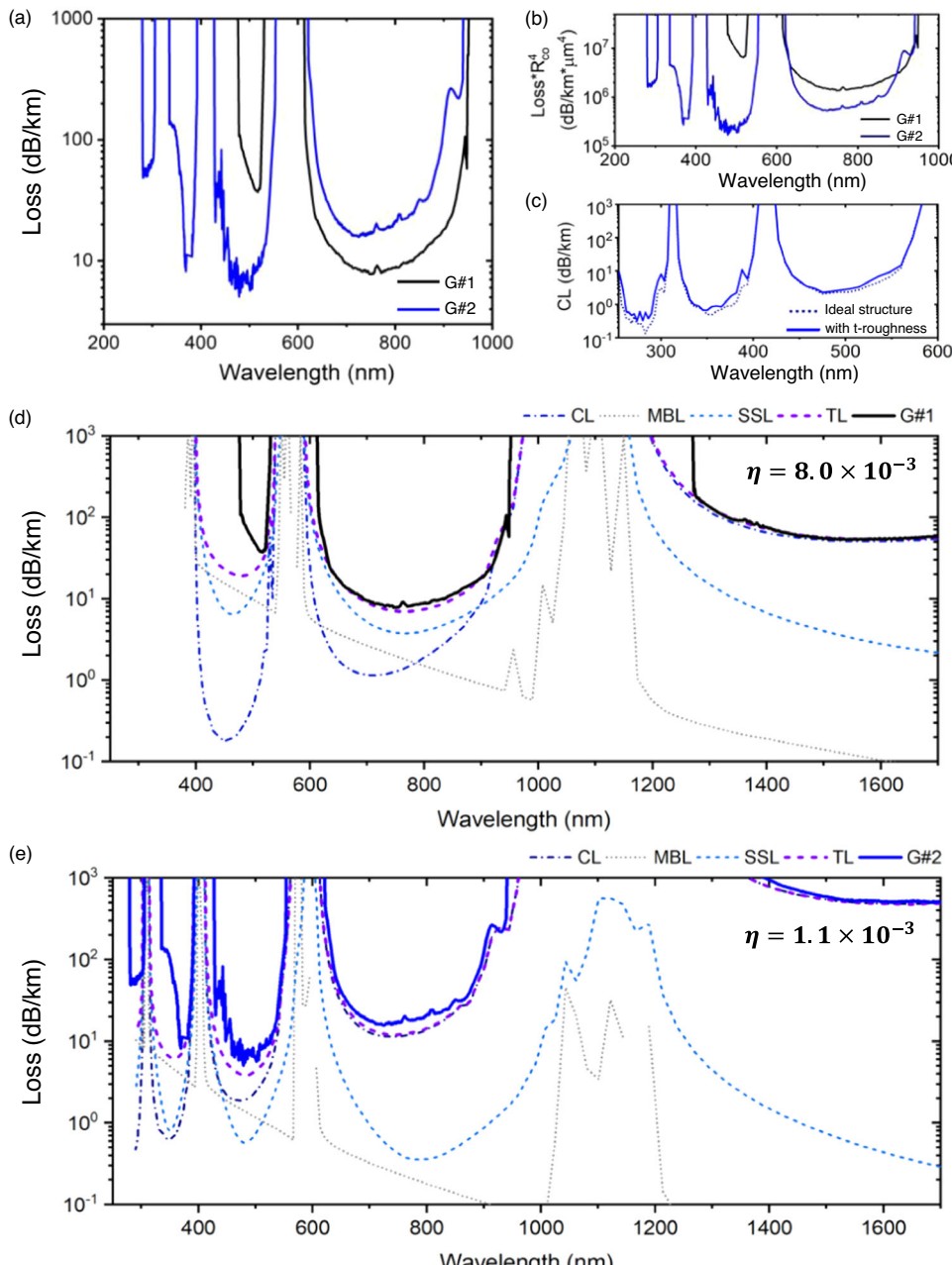

**Fig. 4 | Loss trends comparison and total loss analysis. a** Representative measured loss spectra for fibers in G#1 and G#2 showing the different loss trends in the short-wavelength range. **b** Normalization of the measured fiber loss with respect to the fiber core radius ($R_{co}$). **c** Study of the impact of transverse roughness on the CL. Here the considered SR-TL HCPCF has a core radius of 13.5 µm and a tubes thickness of 0.6 µm. Loss source breakdown for the representative fibers in **d** G#1 and **e** G#2. CL confinement loss, MBL microbending loss, SSL surface scattering loss, TL total loss.

within the fiber optics context. Noteworthily, Fiber A and Fiber B loss figures stand for record low-loss values in the short-wavelength range and lie under the SRSL, which is a fundamental limitation that hinders the reduction of attenuation values of silica-core fibers. In addition, it is remarkable that such ultralow loss figures have been attained by using a fiber architecture as simple as SR-TL HCPCFs without the need for other more complex IC HCPCF structures.

## Discussion

Current developments in visible and ultraviolet photonics call for fibers able to transmit such wavelengths with low loss. Improvements in the loss displayed by solid-core fibers, however, tend to be marginal as they are fundamentally restricted by silica properties. HCPCFs, on the other hand, emerge as promising alternatives to circumvent

silica-core fibers limitations. However, the performances of HCPCF working at short wavelengths have been hitherto limited by scattering processes arising from the roughness of the fiber core surfaces.

In this manuscript, we reported on HCPCFs with reduced core surface roughness obtained by modifying these fibers' usual fabrication techniques. Here, we proposed the use of counter-directional gas and glass flows during the fiber draw to add shear to the fiber silica membranes and, thus, achieve smoother core surfaces. We showed that the rms roughness values were reduced from 0.40 to 0.15 nm via the application of the process proposed herein.

The amelioration of the core surface quality—and, therefore, the reduction of SSL—allowed attaining fibers with record low-loss figures in the short-wavelength range. Remarkably, the record loss figures reported herein have been accomplished by using an HCPCF structure

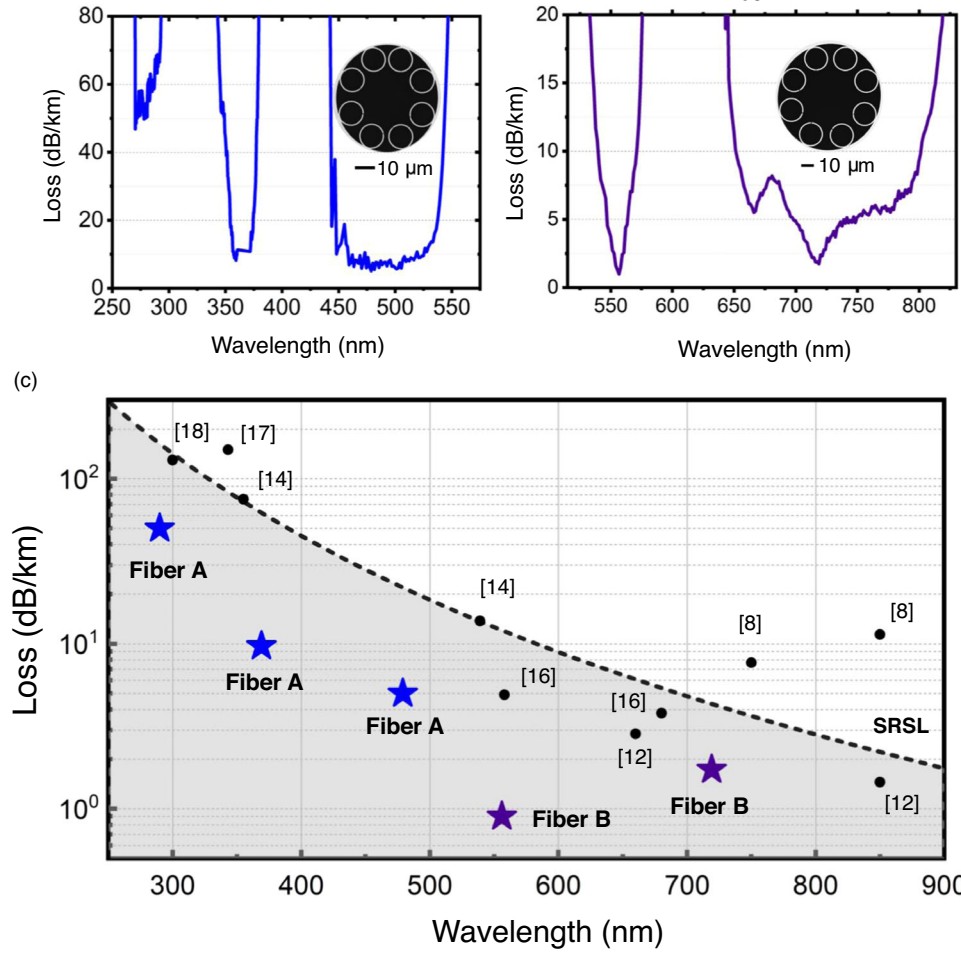

**Fig. 5 | Record low-loss HCPCFs for the short-wavelength range.** Loss measurement results for **a** Fiber A and **b** Fiber B and the fibers' cross-sections. **c** IC HCPCF state-of-the-art framework and the SRSL trend.

as simple as the SR-TL HCPCF design. We understand that the results presented in this paper identify a step-change path for the development of optical fibers operating at short wavelengths. We envisage that our achievements will motivate new research efforts toward further reduction of core surface roughness in HCPCFs and, hence, toward the demonstration of even lower attenuation values, especially in the visible and ultraviolet spectral ranges.

## Methods
### Optical profilometry
Picometer-resolution optical profilometry has been used to assess the height profiles of the HCPCF. The profilometer working principle relies on the reflection and interference of two polarization-modulated laser beams that impinge on spatially-separated sites of the tested sample[22,23]. The optical profilometry measurements have been performed by immersing the fiber under test and by filling the cladding tubes with index-matching liquid to avoid unwelcome reflections.

### Table 2 | Fiber A and Fiber B parameters

| Fiber | $D_{core}$ (µm) | $t$ (µm) | $g$ (µm) | $D_{tubes}$ (µm) |
|---|---|---|---|---|
| A | 27 | 0.6 | 2.1–4.7 | 11 |
| B | 42 | 0.9 | 3.6–5.2 | 18 |

The geometrical parameters of Fiber A and Fiber B.
$D_{core}$ diameter of the core, $t$ tubes' thickness, $g$ gap between the cladding tubes, $D_{tubes}$ diameter of the cladding tubes.

### AFM measurements
A commercial atomic force microscope has been used to characterize the HCPCF core surfaces at high spatial frequency values. For the AFM measurements, the fiber has been angle-cleaved so a clean and debris-free region could be obtained for the realization of the tests.

### Loss measurements
For wavelengths larger than 400 nm, the attenuation values were obtained from cutback measurements using light from a super-continuum source and an optical spectrum analyzer. When accounting for the loss at wavelengths shorter than 400 nm, a plasma lamp was used as the light source, and the transmitted signal was measured in a spectrometer. In the setup for loss measurements in the UV range, since the spectrometer measurements' results are dependent on the alignment of the optical beam directed to it, one placed a flip mirror in-between the fiber end and the spectrometer so to allow correct positioning of the fiber output. The alignment of the fiber output beam with respect to the spectrometer window has, therefore, been performed by using the CCD camera image as a target for beam orientation. When the correct positioning of the fiber end was assured, the mirror was flipped, and the spectrometer measured the transmission signal. During the loss measurements, the fibers have been set in loops of 1 m-bending radius. In addition, we mention that the input coupling conditions have been meticulously adjusted using free-space optics to achieve a fundamental mode-dominated modal content. Also, we carefully inspected the

fibers to identify and remove scattering points arising from fabrication defects and, hence, minimize their impact on the measured fiber loss.

## Data availability

The datasets generated during and/or analyzed during the current study are available from the corresponding author on request.

## Code availability

The code used in the present work is available from the authors upon request. The commercial software COMSOL Multiphysics was used in CL calculations.

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

## Acknowledgements

This research has been funded through PIA program (grant 4F), OzoneFinder project, and la région Nouvelle Aquitaine, and the EU Horizon 2020 grant agreement no. 964531 (F.B.).

## Author contributions

F.B. directed the work. J.H.O., F.A., F.B., F.D., A.D., B.D., and F.Gé. worked on fiber fabrication. J.H.O. performed the optical characterization measurements. J.H.O., A.D., and G.T. performed the profilometry measurements. J.H.O., F.Gé., and F.B. wrote the paper. K.V., F.M., F.Gi., and L.V. worked on simulations. J.H.O., G.T., D.V., and F.B. assessed the surface roughness results. All the authors discussed the results and reviewed the manuscript.

## Competing interests

The authors declare no competing interests.
