## [Peer Review File · Nature Communications]

Hollow-core fibers with reduced surface roughness and ultralow loss in the short-wavelength rangeREVIEWER COMMENTS

Reviewer #1 (Remarks to the Author):

In the present paper, the authors have successfully reduced the core surface roughness of HCPCFs by applying counter-directional glass fluxes within the fiber holes during fiber fabrication. The rms roughness values were reduced from 0.40 nm to 0.15 nm. At the same time, The transmission loss of 50 dB/km at 290 nm, 9.7 dB/km at 369 nm, 5.0 dB/km at 480 nm, and 1.8 dB/km at 719 nm in SR-TL HCPCFs are impressive. These improvements represent an advancement in the State-of-the-Art of this important technology. However, the conceptual novelty is less clear. Instead, as far as we see it, the achievement relies on careful innovative techniques with known principles[17-24], including shear can suppress capillary waves and make interfaces smoother. At the same time, the manuscript also has the following problems:

(1)The introduction is well motivated and many high-impact works of other authors are cited. However, the more representative research progress of the Nested-tubes HCPCFs at 850nm was reported at the OFC conference in 2021. The fabricated nested fiber achieved a 0.6 dB/km loss value at 850 nm.

Ref: H. Sakr et al., "Hollow Core NANFs with Five Nested Tubes and Record Low Loss at 850, 1060, 1300 and 1625nm," 2021 Optical Fiber Communications Conference and Exhibition (OFC), 2021, pp. 1-3.

(2)Although previous studies have suggested that the glass surfaces can retain a structural signature of the direction of the flow that took place during the fiber fabrication, as the most important innovative technique, the authors lack an in-depth mechanism analysis on why counter-directional gas flows can reduce surface roughness.

(3)In this work, a vacuum is applied to the fiber core during the fiber draw to achieve counter-directional gas flows. Whether this approach negatively affects the fiber fabrication process? For example, the stability of the fiber's outer diameter, the longitudinal consistency of the microstructure, etc... What is the length of fiber produced by this method?

(4)The author compares the loss trends of two different SR-TL HCPCFs in Figure 3(b) and concluded that the scattering loss of fiber 2 was smaller. This is not rigorous because the core sizes of the two fibers are very different. In particular, confinement loss, bending loss, and microbending loss are sensitive to core size or core-to-wavelength ratio. These factors will also shift the low-loss band of the fiber with a smaller core size to shorter wavelengths without considering the scattering loss. Therefore, to reach the authors' conclusions, the core diameter of the fiber should be maintained at a similar size.

(5)In this work, Fiber B has an excellent performance in the visible spectral region. However, why is low-loss bandwidth at 719 nm so narrow (shown in Fig. 4(b))? Rather than the overall loss reduction like fiber A around the 500nm transmission band. Does the loss result take into account the impact of the error and uncertainty during the measurement process? The authors should clearly explain the details of the fiber testing process, such as bending radius, length of the test fiber, cutting method, etc., to improve the reliability of fiber loss measurement.

Reviewer #2 (Remarks to the Author):

The manuscript presents an interesting attempt at reducing surface roughness at the air-glass boundary of the core in a hollow-core photonic crystal fiber (HC-PCF). The attempt is interesting, and it would be extremely interesting for the community if it was proven that the approach works, but unfortunately the methods chosen by the authors do not really clearly prove, what they want to prove.

This is a pity, because they could simply have designed their experimental comparisons better, and they might have had convincing proof. In the present form, I sadly must recommend to make major revisions of the paper before the paper should be considered for publication. Below I give details on the problems present in the current manuscript.

1) The major claim of the paper, built on theory and observations in previous papers, is that the surface roughness of the air-glass interface of the core is reduced if one draws the fiber with a gas flow in the cladding tubes, and a counter-propagating flow in the core region. Now, this may be true in itself, but the authors claim that the counter-propagating flow can be achieved simply by applying positive pressure to the cladding tubes and vacuum to the core. But I can't help but wonder: how much gas flow can there actually be through the core during the drawing, considering that in the drawn end of the fiber the core diameter is typically well below 50 μm ? Is there really a sufficient amount of gas moving through the core, and is that enough to provide a sufficient shear stress to dampen the surface capillary waves? Have the authors even tried to quantify these values with simple calculations to support the claim that applying vacuum to the core would provide any noticeable shear stress? How much shear stress is required to reduce the surface capillary waves significantly, how large gas flow is required to reach this shear stress, and is this amount of gas flow realizable through the dimensions of the fiber just by applying vacuum? The authors have not even provided any values for how much vacuum (-5 mbar? -50 mbar? -500 mbar?) they needed to apply to the core to see a reduction in surface roughness, so not a single one of the above extremely important physical properties have been quantified. Although in line 205-206 the authors claim that details "on the fabrication procedures will be provided in the following", there aren't actually any such details given at all.

It is clear that the authors must provide much more of a sound theoretical basis for the claims that simply applying vacuum can result in sufficient gas flow and shear stress to dampen the surface waves.

2) The authors completely neglect that many other parameters have an influence on the surface roughness. As seen in Eqn. (1), the surface roughness will depend on the glass transition temperature, T_G , of the glass. But T_G depends on cooling rate of the glass, so that must mean that the surface roughness will depend on the cooling rate of the glass, which again is determined by drawing parameters such as furnace temperature, drawing speed, and preform diameter (the larger the preform, the larger the mass, which will take longer time to cool down as it exits the furnace). Therefore, one needs to be extremely careful when comparing fibers drawn with and without vacuum in the core. It is clear that all other drawing parameters should be kept as equal as possible, to make a fair comparison, but it is evident that the authors disregarded this and compared fibers made with widely different geometries.

3) The description of previous relevant work in the introduction is mostly accurate and fair, but around lines 74-75 I miss a mention of the relevant work by Pryamikov et al., Opt. Express 19(2) p. 1441 (2010).

4) It is stated in lines 88-92 that according to theory, the surface scattering loss scales quadratically with the surface roughness root-mean-square h_{rms} . Let's say the authors would have simply drawn two fibers with almost identical geometries, but one fiber was made without vacuum in the core (F1), and the other fiber with vacuum in the core (F2). Let's say that the authors then find that F1 has $h_{\text{rms}} = 0.4 \text{ nm}$ and F2 has $h_{\text{rms}} = 0.15$. Then the authors should simply test whether loss (at around the same wavelengths) is lower in F2 by a factor of $(0.15/0.4)^2 = 0.14$. This would be simple and convincing for anyone reading the paper. However, the authors have unfortunately not chosen this approach, and are instead always comparing fibers with very different geometries.

I know that applying vacuum to the core during drawing will change the fiber structure, but this can be compensated for in different ways, for example by keeping the *difference* between core pressure and cladding tube pressure the same, and/or by drawing the fibers from preforms with different initial geometries, so that the fiber structures end up being similar.

5) Line 93-94: "despite recent results show[ing] that adjusting the drawing stress during the fiber fabrication process of fibers diminishes the roughness along the drawing direction [17]". Since this is an interesting way of reducing the surface roughness, please be more specific ("adjusting" is very vague): is there an optimum for the drawing tension? Or should it be as high as possible? And why is this approach not a solution, is it because the required drawing tension is so high that fiber breaks occur too frequently?

6) Line 100: "counter directional glass fluxes", I think you mean "gas fluxes" and not "glass fluxes"?

7) Figure 1(d): there is something strange/wrong in this plot. Looking at the horizontal axis labels, one can see that there is a discontinuity in the horizontal axis at $1 \mu\text{m}^{-1}$, but the blue dashed line is completely straight without any discontinuity, and the black curve showing the data also does not seem to have any discontinuity. Probably there is an error in the horizontal axis labels? The blue dashed line and the black curve should have a discontinuity that looks more like that seen in Fig. 2(d).

8) Lines 142-146 and Fig. 1(d): the authors focus very much on the $1/f$ trend given in Eqn. (1). First, Eqn. (1) is taken from Ref. [2], but no reference is given. Second, Ref. [2] discusses that the more general equation is not only of the form $1/f$ but of the form $(1/f) \times \coth(f \times W/2)$, where W is the perimeter of the cladding tube. This means that it is not correct to only plot a straight line to show the $1/f$ trend in Fig. 1(d) and Fig. 2(d), one also needs to include the \coth -term. In lines 148-149 the authors wonder about the deviation from the $1/f$ behavior for small values of spatial frequency f , but this is actually expected if you don't forget about the $\coth(f \times W/2)$ term.

More importantly: the \coth -term containing the tube perimeter W clearly shows that the strength of the surface capillary waves also depends on the geometry (specifically the diameter) of the cladding tubes! This again therefore shows that one cannot just compare fibers with different geometries, and claim that one fiber has less surface roughness because it was drawn with vacuum in the core. There could be less surface roughness simply because one of the fibers was made with larger cladding tubes (larger W) than the other fiber.

9) In line 148 the authors state that they use $T_G/\gamma = 2000$ (they don't give any units for this value, but it is supposed to be $\text{s}^2 \times \text{K}/\text{kg}$ or $\text{K} \times \text{m}^2/\text{J}$). However, in line 158 they say that typically $T_G = 1500 \text{ K}$ and $\gamma = 0.3 \text{ J}/\text{m}^2$, but using these values (as is also done in Ref. [2]) one actually gets $T_G/\gamma = 5000 \text{ s}^2 \times \text{K}/\text{kg}$, so I don't understand why in line 148 the authors say they use $T_G/\gamma = 2000 \text{ s}^2 \times \text{K}/\text{kg}$?

Very strangely in line 157 the authors state $\sqrt{k_B \times T_G/\gamma} = 0.4 \text{ nm}$. However, if one uses the (in my opinion, more correct) value of $T_G/\gamma = 5000 \text{ s}^2 \times \text{K}/\text{kg}$ the result is actually 0.26 nm . Moreover, even if one uses the value that the authors claim they are using, $T_G/\gamma = 2000 \text{ s}^2 \times \text{K}/\text{kg}$, the result is 0.17 nm . So it is very confusing how the authors arrive at 0.4 nm ? It seems this value is not from a thoughtful calculation, but rather chosen to get better agreement with the experimental measurements in Fig. 2(e) and the horizontal dashed line labelled "TESR"?

10) In line 201 the authors refer to Hagen-Poiseuille's law and how the shear rate "can be correlated with the pressure gradients experienced by the preform components during the fiber drawing", ok, but Hagen-Poiseuille's law relates to pressure drop along a tube, not a transverse pressure gradient which determines, e.g., whether the cladding capillaries expand or contract during the drawing. So it is unclear what the authors want to state here. As mentioned in comment 1), it would be highly relevant if the authors had actually presented some calculations of the shear stress and gas flow inside the core for a given applied vacuum/pressure.

11) Figure 2(e): the rms roughness is plotted versus cladding tube thickness t , why? It would make more sense to plot against cladding tube diameter d or perimeter $W = \pi \times d$, since the roughness spectrum scales with the factor $\coth(f \times W/2)$.

Figure 2(c) and (d) also compare the two groups of fibers G#1 and G#2, but the authors mention

nothing about the geometries of the fibers included in the two groups.

As noted earlier, to be useful a comparison between fibers drawn without and with vacuum in the core would have similar geometrical parameters in both cases, but it is seen in Fig. 2(e) that the authors are comparing fibers with widely different cladding tube thicknesses.

12) In line 231 the authors deduce from Fig. 2(d): "the new fabrication methods reported herein entail a reduction of the PSD values at spatial frequencies lower than $10^{-1} \mu\text{m}^{-1}$, which, in turn, readily impacts the rms roughness of the fibers." Even if we neglect the fact that the comparison in Fig. 2(d) is not relevant unless one compares fibers with similar geometries, the authors should note that although the PSD for G#2 is lower at spatial frequencies $f < 10^{-1} \mu\text{m}^{-1}$, the PSD is seen to be *higher* than for G#1 for spatial frequencies $f > 1 \mu\text{m}^{-1}$! The authors ignore this experimental observation, as if the higher spatial frequencies of roughness are somehow irrelevant for surface scattering. However, considering that as a practical guideline we can say that surface roughness is optically relevant when the roughness is on the scale of the optical wavelength, it becomes interesting to consider that light at 400 nm wavelength should be strongly scattered by surface roughness with a spatial frequency of $1/(400 \text{ nm}) = 2.5 \mu\text{m}^{-1}$. Interestingly, looking at Fig. 2(e), we see that the PSD at a spatial frequency of $2.5 \mu\text{m}^{-1}$ is actually *higher* for G#2 than for G#1! Thus, one would actually expect the surface scattering at a wavelength of 400 nm to be worse for G#2 than for G#1. This is a conclusion completely opposite to that which the authors are making by looking at Fig. 2(d), so the authors should think carefully about their claims and reconsider the statements here.

13) Minor issue: line 234: "In this context, Fig. 1e", here the authors probably intended to refer to Fig. 2(e), not 1(e).

14) It is stated in line 234 that the rms roughness values plotted in Fig. 2(e) actually only account for the optical profilometry measurements, thus implicitly ignoring the AFM measurements at higher spatial frequencies. This is actually a significant problem with this comparison, because we know from Fig. 2(d) that the PSD is actually higher for G#2 than for G#1 precisely in the spatial frequency range ($f > 1 \mu\text{m}^{-1}$) where only the AFM measurements provide information. It is simply not reasonable to simply leave out spatial frequencies above $1 \mu\text{m}^{-1}$ and then claim that G#2 has lower rms roughness, especially when we know from comment 12) that it is precisely in this spatial frequency range that the roughness is relevant for the optical wavelengths of interest.

15) Figure 3(b): the comparison here is also meaningless, since G#1 has a much larger core than G#2. One could suspect that both fibers were made from the same preform, but that the cladding tubes in G#2 were made larger by applying vacuum in the core (also making the core smaller). If G#2 has larger cladding tubes, then the larger perimeter W is expected to lead to lower surface roughness, thus making the comparison of no value to support the authors' claim that larger shear forces have lowered the surface roughness.

One can also explain the loss difference between G#1 and G#2 in another way completely unrelated to surface roughness. As the authors state themselves, the lower loss of G#1 between 600 nm and 1000 nm is likely due just to the larger core diameter of G#1. Now, the authors claim that the lower loss of G#2 compared to G#1 around 500 nm is allegedly due to lower surface roughness at shorter wavelengths. However, it is also well known that the critical radius at which bend loss occurs scales as D^3/λ^2 (see e.g. Frosz et al., *Photonics Research*, 5(2) p. 88 (2017)). Note that smaller critical radius implies greater resilience against bend loss. This means that a fiber with smaller core (such as G#2) can be expected to be more resistant against bend loss than a fiber with larger core (such as G#1), especially at shorter wavelengths. The fact that G#2 has lower loss than G#1 at shorter wavelengths could therefore be due just to smaller bend loss, and it is clear that comparing fibers with significantly different core diameters, as is done in Fig. 3(b), is a very bad idea when one is trying to prove that there is a difference due to surface scattering loss. Other geometrical differences between G#1 and G#2 could also play a role, but the authors provide no information about, for example, the cladding tube diameters or the gap size between them. So the statement in line 277 ("The distinction between the loss behaviors of fiber in G#1 and G#2 is, therefore, the optical

manifestation of the SSL reduction in G#2 fibers") is clearly a claim without any scientific foundation at all.

16) Figure 4(c): the dashed line showing the *solid-core* silica Rayleigh scattering limit (SRSL) is misleading to be included in this way when comparing with loss in *hollow-core* fibers. If including such a comparison of loss in hollow-core fibers with the SRSL, it would only be fair to multiply the SRSL with the field-glass overlap ($\ll 1\%$) to get closer to the actual loss that can be expected in a hollow-core fiber from scattering in the glass.

It would be more correct to instead compare with the surface scattering loss (SSL) as shown for example in Fig. 3(a). Why did the authors not instead plot the SSL curve from Fig. 3(a) in Fig. 4(c)? That would be a much more logical and fair comparison.

Interestingly, when considering the theoretical total loss (TL) in Fig. 3(a), it is actually considerably lower than all experimentally achieved loss values here ("Fiber A" and Fiber "B"), which is actually a good indication that the scattering factor η (set to either 300 or 2100 in Fig. 3(a)) was not significantly lowered in Fiber A and Fiber B, despite the attempt to increase shear stress by applying vacuum to the core.

17) Another problem with the comparison in Fig. 4(c) between earlier work and "Fiber A" and "Fiber B": how can the authors be sure that none of the fibers presented in the earlier work, were also not drawn while applying vacuum to the core? I could easily imagine that some of the earlier work also applied vacuum to the core, but that the fibers had higher loss for other reasons (for example, different core diameter, bend loss, etc.).

18) Line 334-335: "In the setup for loss measurements in the UV range, one placed a flip mirror in-between the fiber end and the spectrometer so to allow correct positioning of the fiber output by observing its image in a CCD camera." This description is somewhat confusing, don't you rather mean that you observed the output on the CCD camera to ensure that you maximized coupling into the core at the *input* end of the fiber? The "correct positioning of the fiber output" is confusing without more explanation.

In conclusion, the fibers presented here have interestingly low loss in the UV-VIS range, but there is no strong evidence for what the authors are trying to claim: that applying vacuum to the core during fiber drawing will lead to lower surface roughness and thereby lower loss. The comparisons between fibers cannot be made using fibers with significantly different geometries, because the impact of the geometry on confinement loss and bend loss, not to mention even on the surface roughness through the perimeter W , cannot be ignored. Even the drawing conditions (furnace temperature, drawing speed, etc.) will change T_G and thus the surface roughness.

Answer to the reviewers

We acknowledge the reviewers for assessing our manuscript and for the pertinent comments. In the following, we provide the answers to the reviewers' comments. We hope that our clarifications adequately address the issues raised by the reviewers.

Reviewer #1 (Remarks to the Author):

In the present paper, the authors have successfully reduced the core surface roughness of HCPCFs by applying counter-directional glass fluxes within the fiber holes during fiber fabrication. The rms roughness values were reduced from 0.40 nm to 0.15 nm. At the same time, The transmission loss of 50 dB/km at 290 nm, 9.7 dB/km at 369 nm, 5.0 dB/km at 480 nm, and 1.8 dB/km at 719 nm in SR-TL HCPCFs are impressive. These improvements represent an advancement in the State-of-the-Art of this important technology. However, the conceptual novelty is less clear. Instead, as far as we see it, the achievement relies on careful innovative techniques with known principles [17-24], including shear can suppress capillary waves and make interfaces smoother. At the same time, the manuscript also has the following problems:

We thank the reviewer for the appreciation of our work. We have made amendments to the manuscript text as per the reviewer's recommendations. Our comments on the reviewer's concerns can be found below.

(1) The introduction is well motivated and many high-impact works of other authors are cited. However, the more representative research progress of the Nested-tubes HCPCFs at 850nm was reported at the OFC conference in 2021. The fabricated nested fiber achieved a 0.6 dB/km loss value at 850 nm.

Ref: H. Sakr et al., "Hollow Core NANFs with Five Nested Tubes and Record Low Loss at 850, 1060, 1300 and 1625nm," 2021 Optical Fiber Communications Conference and Exhibition (OFC), 2021, pp. 1-3.

In attention to the reviewer's comments, we included the suggested reference in the manuscript text. Now it reads:

"Nested-tubes HCPCFs, in turn, display loss figures of 0.6 dB/km at 850 nm and 2.85 dB/km at 660 nm [12, 15]."

(2) Although previous studies have suggested that the glass surfaces can retain a structural signature of the direction of the flow that took place during the fiber fabrication, as the most important innovative technique, the authors lack an in-depth mechanism analysis on why counter-directional gas flows can reduce surface roughness.

We agree with the reviewer that this part wasn't developed in the original manuscript. Our thinking then is that adding a section detailing the rheology dynamics would impact the

fluency and the unity of message of the manuscript. In light of your comment and that of the second reviewer, we believe that it would make sense to develop on how the shear takes place.

In the new version of the manuscript, we have included, within the section *“Shear stress as means to structure the surface roughness profile”*, a simplified model which allows accounting for the increase in the effective interfacial tension via the estimation of the shear rate inside the fiber core. This, in turn, allows having an estimation of the mean square roughness reduction factor due to the increase in the effective interfacial tension.

(3) In this work, a vacuum is applied to the fiber core during the fiber draw to achieve counter-directional gas flows. Whether this approach negatively affects the fiber fabrication process? For example, the stability of the fiber's outer diameter, the longitudinal consistency of the microstructure, etc... What is the length of fiber produced by this method?

Our experience showed that applying vacuum in the fiber core during fabrication conferred a highly stable fiber drawing, with outer diameter variations no larger than 0.5%. Also, evaluation of the microstructure elements allowed for ascertaining a highly consistent fiber cross-section along the 500-1000 m typically fabricated lengths. This information can now be found in the manuscript text in the following excerpt:

“During the fabrication routines following our new methods, the fiber outer diameter variation was no larger than 0.5%. Also, evaluation of the fiber microstructure elements allowed ascertaining a highly consistent fiber cross-section along the 500-1000m typically fabricated lengths.”

Moreover, in the revised manuscript we emphasized in the section related to the shear flow model on the requirements for establishing a shear rate whilst keeping the aimed structural integrity of the fiber.

(4) The author compares the loss trends of two different SR-TL HCPCFs in Figure 3(b) and concluded that the scattering loss of fiber 2 was smaller. This is not rigorous because the core sizes of the two fibers are very different. In particular, confinement loss, bending loss, and microbending loss are sensitive to core size or core-to-wavelength ratio. These factors will also shift the low-loss band of the fiber with a smaller core size to shorter wavelengths without considering the scattering loss. Therefore, to reach the authors' conclusions, the core diameter of the fiber should be maintained at a similar size.

In the new version of the manuscript, we have added new graphs in Fig. 4 (formerly Fig. 3) to clarify that, although the core diameters of the fibers are different, one can correlate the difference between the measured loss trends to the diminishment of the surface scattering loss in the fibers and, hence, to the amelioration of the quality of the core surfaces of the fibers fabricated by our new methods. In Fig. 4b, we now show a plot in which we multiply the loss figures shown in Fig. 4a by R_{co}^4 , where R_{co} is the radius of the fiber core. Multiplication by such a factor allows normalizing the loss with respect to the core size, as studied in the scaling laws available in reference [31] in the manuscript text. We observe that the loss normalization shown in Fig. 4b further clarifies the difference between the loss trends of the fibers in G#1 and G#1.

Additionally, in attention to the reviewer's comments, we included in Fig. 4d and Fig. 4e simulations regarding the confinement loss (with integrated bending loss) and microbending loss. The data in the new figure ascertains that the difference between the loss behaviors can

be correlated to the surface scattering loss decrease and that such a decrease is consistent with the reduction of the measured rms surface roughness we report in the manuscript.

(5) In this work, Fiber B has an excellent performance in the visible spectral region. However, why is low-loss bandwidth at 719 nm so narrow (shown in Fig. 4(b))? Rather than the overall loss reduction like fiber A around the 500nm transmission band. Does the loss result take into account the impact of the error and uncertainty during the measurement process? The authors should clearly explain the details of the fiber testing process, such as bending radius, length of the test fiber, cutting method, etc., to improve the reliability of fiber loss measurement.

Yes, we have been very careful when conducting the measurements on the transmission loss of the fibers. In the loss measurements, the fibers have been set in loops of 1 m-bending radius, and the typical 250 m-long fiber pieces used in our cutback measurements afforded suitable power variation for the fiber loss calculations. Also, we mention that fiber cleaving has been very carefully performed and that the input coupling conditions have been meticulously adjusted using free-space optics so the modal content of the fiber could be fundamental mode-dominated. Regarding the loss spectrum of Fiber B, we consider that its relatively narrow bandwidth around 719 nm is likely to be due to scattering points arising from slight fabrication defects in the tested fiber. In our measurements, we endeavored to select fiber pieces that displayed minimum occurrences of scattering points (by visually inspecting and removing the faulty parts). However, a number of weak scattering points are likely to remain in the tested fiber pieces and affect their transmission spectral characteristics. In the new version of the paper, we have included in the “Materials and Methods” section (“Loss measurements” subsection) additional information on the loss measurement procedure, as per the reviewer’s suggestions.

Reviewer #2 (Remarks to the Author):

The manuscript presents an interesting attempt at reducing surface roughness at the air-glass boundary of the core in a hollow-core photonic crystal fiber (HC-PCF). The attempt is interesting, and it would be extremely interesting for the community if it was proven that the approach works, but unfortunately the methods chosen by the authors do not really clearly prove, what they want to prove. This is a pity, because they could simply have designed their experimental comparisons better, and they might have had convincing proof. In the present form, I sadly must recommend to make major revisions of the paper before the paper should be considered for publication. Below I give details on the problems present in the current manuscript.

We thank the reviewer for the appreciation of our work. We have made amendments to the manuscript text as per the reviewer’s recommendations. Our comments on the reviewer’s concerns can be found below.

1) The major claim of the paper, built on theory and observations in previous papers, is that the surface roughness of the air-glass interface of the core is reduced if one draws the fiber with a gas flow in the cladding tubes, and a counter-propagating flow in the core region. Now, this may be true in itself, but the authors claim that the counter-propagating flow can be achieved simply

by applying positive pressure to the cladding tubes and vacuum to the core. But I can't help but wonder: how much gas flow can there actually be through the core during the drawing, considering that in the drawn end of the fiber the core diameter is typically well below 50 μm ? Is there really a sufficient amount of gas moving through the core, and is that enough to provide a sufficient shear stress to dampen the surface capillary waves? Have the authors even tried to quantify these values with simple calculations to support the claim that applying vacuum to the core would provide any noticeable shear stress? How much shear stress is required to reduce the surface capillary waves significantly, how large gas flow is required to reach this shear stress, and is this amount of gas flow realizable through the dimensions of the fiber just by applying vacuum? The authors have not even provided any values for how much vacuum (-5 mbar? -50 mbar? -500 mbar?) they needed to apply to the core to see a reduction in surface roughness, so not a single one of the above extremely important physical properties have been quantified. Although in line 205-206 the authors claim that details "on the fabrication procedures will be provided in the following", there aren't actually any such details given at all. It is clear that the authors must provide much more of a sound theoretical basis for the claims that simply applying vacuum can result in sufficient gas flow and shear stress to dampen the surface waves.

In consideration of the reviewer's concerns, we have incorporated, in the new version of the manuscript, within the section "*Shear stress as means to structure the surface roughness profile*", a simplified model which allows accounting for the increase in the effective interfacial tension via the estimation of the shear rate inside the fiber core. The model reported in the new version of the paper allows attaining an estimation of the mean square roughness reduction factor due to the increase in the effective interfacial tension. Even though it consists of a simplified model that does not consider all the complexity of the fiber drawing procedure, it has the merit of satisfactorily demonstrate the requirement for the establishment of a shear flow; it encapsulates the main conceptual items in designing further fabrication processes for surface-roughness suppression in fiber drawing, by relating the shear rate applied to the fiber microstructure with the pressure gradients and the draw down-ratio applied during fabrication. We hope that the model and discussions available in the new version of the manuscript adequately address the reviewer's concerns.

2) The authors completely neglect that many other parameters have an influence on the surface roughness. As seen in Eqn. (1), the surface roughness will depend on the glass transition temperature, T_G , of the glass. But T_G depends on cooling rate of the glass, so that must mean that the surface roughness will depend on the cooling rate of the glass, which again is determined by drawing parameters such as furnace temperature, drawing speed, and preform diameter (the larger the preform, the larger the mass, which will take longer time to cool down as it exits the furnace). Therefore, one needs to be extremely careful when comparing fibers drawn with and without vacuum in the core. It is clear that all other drawing parameters should be kept as equal as possible, to make a fair comparison, but it is evident that the authors disregarded this and compared fibers made with widely different geometries.

We regret that the previous version of our manuscript has not been sufficiently clear in describing the diligence in setting up the fiber drawing parameters and the rigorousness of our analyses. We clarify that by first stating that we are aware that the characteristics of the core surface roughness in the drawn fibers are dependent on several parameters and that the fiber drawing procedure has a multifactorial character. Hence, to adequately analyze our results, we have endeavored to have, in G#1 and G#2, fibers with comparable geometrical dimensions. This information can now be verified in Table 1, which has been added to the new version of the

paper. Additionally, we have added the following fragment to the text to clarify the point raised by the reviewer:

“Table 1 presents the geometrical dimensions of the fibers in G#1 and G#2. It is worth mentioning that, to adequately perform our analyses, we have endeavored to have, in G#1 and G#2, fibers with comparable cladding tubes dimensions, as one can verify in Table 1.”

3) The description of previous relevant work in the introduction is mostly accurate and fair, but around lines 74-75 I miss a mention of the relevant work by Pryamikov et al., Opt. Express 19(2) p. 1441 (2010).

We agree with the reviewer on the significance of the reference, and we included the following comment in the manuscript text to mention the work by Pryamikov et al:

“Also, the demonstration of guidance in single-ring tubular lattice (SR-TL) HCPCFs [7] stands out as an important achievement that motivated further developments in HCPCF technology.”

4) It is stated in lines 88-92 that according to theory, the surface scattering loss scales quadratically with the surface roughness root-mean-square h_{rms} . Let's say the authors would have simply drawn two fibers with almost identical geometries, but one fiber was made without vacuum in the core (F1), and the other fiber with vacuum in the core (F2). Let's say that the authors then find that F1 has $h_{rms} = 0.4$ nm and F2 has $h_{rms} = 0.15$. Then the authors should simply test whether loss (at around the same wavelengths) is lower in F2 by a factor of $(0.15/0.4)^2 = 0.14$. This would be simple and convincing for anyone reading the paper. However, the authors have unfortunately not chosen this approach, and are instead always comparing fibers with very different geometries. I know that applying vacuum to the core during drawing will change the fiber structure, but this can be compensated for in different ways, for example by keeping the *difference* between core pressure and cladding tube pressure the same, and/or by drawing the fibers from preforms with different initial geometries, so that the fiber structures end up being similar.

As the measurement of the transmission loss of the fiber integrates different sources of loss in addition to scattering points arising from slight fabrication defects, estimating the reduction of the surface scattering loss is unfortunately not as straightforward as formulated in the question. To address the reviewer's concern, we adopted the approach of presenting, in the new version of the manuscript, the simulated loss for both the fibers studied in Fig. 4 considering different loss mechanisms (as now appears in Fig. 4d and Fig. 4e). The data in the new figure ascertains that the difference between the loss behaviors can be correlated to the surface scattering loss decrease and that such a decrease is consistent with the reduction of the measured rms surface roughness we report in the manuscript. This information can now be found in the following fragment:

“The most striking information one can assimilate from Fig. 4d and Fig. 4e concerns the SSL. Data in Fig. 4d (G#1 representative fiber) shows that the factor η in the SSL expression $(\alpha_{SSL} = \eta \times F \times (\lambda/\lambda_0)^{-3}$, as described in the introduction) that entails adequate fitting between the simulated TL and the measured loss is $\eta_{G\#1} = 8.0 \times 10^{-3}$. In turn, the value of η that allows attaining suitable fitting between the simulated TL and the measured loss for the representative fiber in G#2 is $\eta_{G\#2} = 1.1 \times 10^{-3}$. If one reminds that, as discussed in the introduction, η can be related to the square of the rms surface roughness, the ratio between

the square roots of the obtained η can be readily associated with the reduction of the rms core surface roughness values in our fibers. Remarkably, $\sqrt{\frac{\eta_{G\#1}}{\eta_{G\#2}}} \approx 2.7$, which is consistent with our surface roughness measurement results.

5) Line 93-94: “despite recent results show[ing] that adjusting the drawing stress during the fiber fabrication process of fibers diminishes the roughness along the drawing direction [17]”. Since this is an interesting way of reducing the surface roughness, please be more specific (“adjusting” is very vague): is there an optimum for the drawing tension? Or should it be as high as possible? And why is this approach not a solution, is it because the required drawing tension is so high that fiber breaks occur too frequently?

Yes, the results reported in [19] (new numbering of [17] in the new version of the paper), show that increasing the drawing stress to larger levels implies reducing the surface roughness height along the fiber axis. However, the reviewer is correct when mentioning that such an approach is constrained to acceptable drawing tensions that do not entail frequent breaking of the fiber during fabrication. In attention to the reviewer's comments, we have added more specific comments to the fragment referenced in the reviewer's question:

“In this context, despite recent results showing that increasing the drawing stress during the fabrication process of silica capillaries diminishes the roughness along the drawing direction [19], no work has been reported so far on HCPCFs and on how to mitigate SSL-dominated scenarios.”

6) Line 100: “counter directional glass fluxes”, I think you mean “gas fluxes” and not “glass fluxes”?

We thank the reviewer for pointing out our mistake. It is now fixed in the new version of the manuscript.

7) Figure 1(d): there is something strange/wrong in this plot. Looking at the horizontal axis labels, one can see that there is a discontinuity in the horizontal axis at $1 \mu\text{m}^{-1}$, but the blue dashed line is completely straight without any discontinuity, and the black curve showing the data also does not seem to have any discontinuity. Probably there is an error in the horizontal axis labels? The blue dashed line and the black curve should have a discontinuity that looks more like that seen in Fig. 2(d).

We clarify that the data in Fig. 1d is correct. Our impression is that our choice of using different scale breaks in the vertical axis of the plots in Fig. 1d and Fig. 2d (Fig. 3d in the new version of the manuscript) has unfortunately generated confusion. To avoid any misunderstanding on the presented data, we now present the above-mentioned graphs with the same scale breaks in the vertical axis.

8) Lines 142-146 and Fig. 1(d): the authors focus very much on the $1/f$ trend given in Eqn. (1). First, Eqn. (1) is taken from Ref. [2], but no reference is given. Second, Ref. [2] discusses that the more general equation is not only of the form $1/f$ but of the form $(1/f) \times \coth(f \times W/2)$, where W

is the perimeter of the cladding tube. This means that it is not correct to only plot a straight line to show the $1/f$ trend in Fig. 1(d) and Fig. 2(d), one also needs to include the coth-term. In lines 148-149 the authors wonder about the deviation from the $1/f$ behavior for small values of spatial frequency f , but this is actually expected if you don't forget about the $\coth(f \times W/2)$ term. More importantly: the coth-term containing the tube perimeter W clearly shows that the strength of the surface capillary waves also depends on the geometry (specifically the diameter) of the cladding tubes! This again therefore shows that one cannot just compare fibers with different geometries, and claim that one fiber has less surface roughness because it was drawn with vacuum in the core. There could be less surface roughness simply because one of the fibers was made with larger cladding tubes (larger W) than the other fiber.

In attention to the reviewer's comments, we have now added the PSD trend modulated by the hyperbolic cosine term in our plots and added the comment reproduced below to the manuscript text. We clarify that we are aware of the reliance of the surface capillary waves dynamics on the fiber geometry and, thus, have used fibers with similar tubes sizes in our analyzes, as the reviewer can now observe in Table 1, which has been added in the new version of the manuscript.

“As an additional comparison, we show in Fig. 1d a modified trend (green dashed line in Fig. 1d, attained when quantizing the SCW transversally), obtained by multiplying Eq. (1) by $\cosh\left(\frac{Wf}{2}\right)$, where W is the perimeter of the cladding tubes [2]. The latter is calculated here for cladding tubes with 15 μm diameter, a typical value in our fibers. While the trend modulated by the hyperbolic cosine term allows calculating larger PSD values at low spatial frequencies compared to the $1/f$ trend, the measured PSD remains larger than it, consistently with what has been reported previously [23].”

9) In line 148 the authors state that they use $T_G/\gamma = 2000$ (they don't give any units for this value, but it is supposed to be $\text{s}^2 \times \text{K}/\text{kg}$ or $\text{K} \times \text{m}^2/\text{J}$). However, in line 158 they say that typically $T_G = 1500 \text{ K}$ and $\gamma = 0.3 \text{ J}/\text{m}^2$, but using these values (as is also done in Ref. [2]) one actually gets $T_G/\gamma = 5000 \text{ s}^2 \times \text{K}/\text{kg}$, so I don't understand why in line 148 the authors say they use $T_G/\gamma = 2000 \text{ s}^2 \times \text{K}/\text{kg}$? Very strangely in line 157 the authors state $\sqrt{k_B \times T_G/\gamma} = 0.4 \text{ nm}$. However, if one uses the (in my opinion, more correct) value of $T_G/\gamma = 5000 \text{ s}^2 \times \text{K}/\text{kg}$ the result is actually 0.26 nm . Moreover, even if one uses the value that the authors claim they are using, $T_G/\gamma = 2000 \text{ s}^2 \times \text{K}/\text{kg}$, the result is 0.17 nm . So it is very confusing how the authors arrive at 0.4 nm ? It seems this value is not from a thoughtful calculation, but rather chosen to get better agreement with the experimental measurements in Fig. 2(e) and the horizontal dashed line labelled “TESR”?

We regret that the previous version of our text generated confusion. Firstly, we mention that we have added the missing units in T_G/γ values. Regarding the use of $T_G/\gamma = 2000 \text{ K}\cdot\text{m}^2/\text{J}$ to account for the PSD trends in Fig. 1d, we clarify that it was due to the fact that, as mentioned by Roberts *et al.* (reference [2] in the manuscript text), measurements on the surface tension of drawn capillaries have shown a value of $0.7 \text{ J}/\text{m}^2$, somewhat greater than $0.3 \text{ J}/\text{m}^2$ value, commonly assigned to silica. Using $0.7 \text{ J}/\text{m}^2$ as the surface tension readily gives a T_G/γ in the order of $2000 \text{ K}\cdot\text{m}^2/\text{J}$. We deem that using this value in our PSD plots is reasonable as it portrays the scenario of typical hollow-core fiber fabrication procedures. To address this issue, we added the following comment in the manuscript text:

“Here, we used $T_G/\gamma = 2000 \text{ K.m}^2/\text{J}$, which is consistent with the T_G and γ values reported in [2] for drawn silica capillaries.”

On the other hand, we consider that using the 0.4 nm rms roughness level to contextualize our results allows the reader to compare our results with an established benchmark in silica surface roughness studies. Although part of the literature does not explicitly inform it, the 0.4 nm value is attained when setting upper and lower spatial cutoffs in the calculations. To be more accurate in our writings, we have added the complete expression for calculating the 0.4 nm rms roughness level. It can now be found, in the new version of the manuscript, in the following excerpt:

“In the absence of shear, the rms height of these frozen fluctuations amounts to $\sqrt{\left(\frac{k_B T_G}{2\pi\gamma}\right) \ln\left(\frac{\Lambda_M}{\Lambda_m}\right)} \approx 0.4 \text{ nm}$, which is obtained when using $T_G = 1500 \text{ K}$, $\gamma \approx 0.3 \text{ J/m}^2$, and, for the upper and lower spatial cutoffs, Λ_M and Λ_m , associated respectively with the silica capillary length (4mm) and molecular length (0.5 nm) the values of 4 mm and 0.5 nm, respectively [25]. This rms height level is, hence, referenced as thermodynamic equilibrium surface roughness (TESR). As the reader will observe in the following, here we use such a level as a benchmark to contextualize our results.”

10) In line 201 the authors refer to Hagen-Poiseuille’s law and how the shear rate “can be correlated with the pressure gradients experienced by the preform components during the fiber drawing”, ok, but Hagen-Poiseuille’s law relates to pressure drop along a tube, not a transverse pressure gradient which determines, e.g., whether the cladding capillaries expand or contract during the drawing. So it is unclear what the authors want to state here. As mentioned in comment 1), it would be highly relevant if the authors had actually presented some calculations of the shear stress and gas flow inside the core for a given applied vacuum/pressure.

As mentioned in the answer to the reviewer’s first comment, in the new version of the manuscript, we provide a model which allows accounting for the increase in the effective interfacial tension via the estimation of the shear rate inside the fiber core and, thus, estimating the mean square roughness reduction factor due to the increase in the effective interfacial tension. In the revised manuscript, the description of the used model substituted the former mention of Hagen-Poiseuille’s law in the previous version of the manuscript. Additionally, we clarify that, although the transverse expansion of the cladding tubes has been very carefully tuned in our fiber drawings so as to attain fibers with optimized optical performances, in the present work, we are concerned about the axial dynamics of the surface and the corresponding surface roughness along the fiber axis. Finally, we believe that the revised manuscript addresses the reviewer’s sound point on whether there is a gas flow under HCPCF drawing conditions.

11) Figure 2(e): the rms roughness is plotted versus cladding tube thickness t , why? It would make more sense to plot against cladding tube diameter d or perimeter $W=\pi \times d$, since the roughness spectrum scales with the factor $\coth(f \times W/2)$. Figure 2(c) and (d) also compare the two groups of fibers G#1 and G#2, but the authors mention nothing about the geometries of the fibers included in the two groups. As noted earlier, to be useful a comparison between fibers drawn without and with vacuum in the core would have similar geometrical parameters in both cases, but it is seen in Fig. 2(e) that the authors are comparing fibers with widely different cladding tube thicknesses.

In attention to the reviewer's comment, we have added, in the new version of the manuscript, a new table (Table 1) displaying the geometrical parameters of the fibers we considered in our investigation. This new table allows the reader to verify that the fibers in G#1 and G#2 have tube diameters lying within similar intervals and that our analyzes are consistent. While the rms core roughness data shown in Fig. 3e (previously Fig. 2e) do not show obvious trends with respect to thickness nor to cladding tubes' diameters, showing the tubes' thickness values in the horizontal axis has been simply a choice of presentation. Independently from that choice, the message we would like to convey is that, by using our new fabrication methods (G#2 fibers), we could accomplish fibers with lower core surface roughness than in fibers fabricated by our previous techniques (G#1 fibers). Thus, to avoid any misleadingness, the horizontal axis of Fig. 3e in the new version of the manuscript now shows the sample numbering as they appear in Table 1.

12) In line 231 the authors deduce from Fig. 2(d): "the new fabrication methods reported herein entail a reduction of the PSD values at spatial frequencies lower than $10^{(-1)} \mu\text{m}^{(-1)}$, which, in turn, readily impacts the rms roughness of the fibers." Even if we neglect the fact that the comparison in Fig. 2(d) is not relevant unless one compares fibers with similar geometries, the authors should note that although the PSD for G#2 is lower at spatial frequencies $f < 10^{(-1)} \mu\text{m}^{(-1)}$, the PSD is seen to be *higher* than for G#1 for spatial frequencies $f > 1 \mu\text{m}^{(-1)}$! The authors ignore this experimental observation, as if the higher spatial frequencies of roughness are somehow irrelevant for surface scattering. However, considering that as a practical guideline we can say that surface roughness is optically relevant when the roughness is on the scale of the optical wavelength, it becomes interesting to consider that light at 400 nm wavelength should be strongly scattered by surface roughness with a spatial frequency of $1/(400 \text{ nm}) = 2.5 \mu\text{m}^{(-1)}$. Interestingly, looking at Fig. 2(e), we see that the PSD at a spatial frequency of $2.5 \mu\text{m}^{(-1)}$ is actually *higher* for G#2 than for G#1! Thus, one would actually expect the surface scattering at a wavelength of 400 nm to be worse for G#2 than for G#1. This is a conclusion completely opposite to that which the authors are making by looking at Fig. 2(d), so the authors should think carefully about their claims and reconsider the statements here.

The reason behind our statement that the reduction of the PSD at low spatial frequencies readily impacts the rms roughness of the fibers is that the PSD values at lower spatial frequencies are orders of magnitude greater than the corresponding figures at larger frequencies. For example, for fibers in G#1, the PSD values at frequencies between $10^{-2} \mu\text{m}^{-1}$ and $10^{-1} \mu\text{m}^{-1}$ can reach values as high as $\sim 10 \text{ nm}^2/\mu\text{m}^{-1}$, while the PSD for spatial frequencies between $1 \mu\text{m}^{-1}$ and $10 \mu\text{m}^{-1}$ lies within 10^{-4} to $10^{-2} \text{ nm}^2/\mu\text{m}^{-1}$ interval. The noteworthy difference in the PSD values for low and high spatial frequencies allows concluding that the former has a stronger impact on the rms roughness values than the latter. In the revised manuscript, we commented on this aspect in the following fragment:

"It is worth mentioning that, although one observes larger PSD values at spectral frequencies greater than $1 \mu\text{m}^{-1}$ for fibers in G#2 than for fibers in G#1, the PSD figures at larger spatial frequencies are orders of magnitude lower than the corresponding PSD values at lower spatial frequencies. It entails that the spectral components of the roughness at higher frequencies have a significantly reduced impact on the rms surface roughness compared with the components at lower spatial frequencies."

13) Minor issue: line 234: “In this context, Fig. 1e”, here the authors probably intended to refer to Fig. 2(e), not 1(e).

We thank the reviewer for pointing out our mistake. It is now fixed in the new version of the manuscript.

14) It is stated in line 234 that the rms roughness values plotted in Fig. 2(e) actually only account for the optical profilometry measurements, thus implicitly ignoring the AFM measurements at higher spatial frequencies. This is actually a significant problem with this comparison, because we know from Fig. 2(d) that the PSD is actually higher for G#2 than for G#1 precisely in the spatial frequency range ($f > 1 \mu\text{m}^{-1}$) where only the AFM measurements provide information. It is simply not reasonable to simply leave out spatial frequencies above $1 \mu\text{m}^{-1}$ and then claim that G#2 has lower rms roughness, especially when we know from comment 12) that it is precisely in this spatial frequency range that the roughness is relevant for the optical wavelengths of interest.

As we have mentioned in the answer to question #12, the PSD values at lower spatial frequencies are orders of magnitude larger than the corresponding figures at larger frequencies. Thus, such a great difference between the PSD values for low and high spatial frequencies entails that the former has a stronger influence on the rms roughness values than the latter. Hence, we consider that the data in Fig. 3e (formerly Fig. 2e) are representative of the surface roughness reduction when comparing G#1 and G#2 fiber groups.

15) Figure 3(b): the comparison here is also meaningless, since G#1 has a much larger core than G#2. One could suspect that both fibers were made from the same preform, but that the cladding tubes in G#2 were made larger by applying vacuum in the core (also making the core smaller). If G#2 has larger cladding tubes, then the larger perimeter W is expected to lead to lower surface roughness, thus making the comparison of no value to support the authors' claim that larger shear forces have lowered the surface roughness. One can also explain the loss difference between G#1 and G#2 in another way completely unrelated to surface roughness. As the authors state themselves, the lower loss of G#1 between 600 nm and 1000 nm is likely due just to the larger core diameter of G#1. Now, the authors claim that the lower loss of G#2 compared to G#1 around 500 nm is allegedly due to lower surface roughness at shorter wavelengths. However, it is also well known that the critical radius at which bend loss occurs scales as D^3/λ^2 (see e.g. Frosz et al., *Photonics Research*, 5(2) p. 88 (2017)). Note that smaller critical radius implies greater resilience against bend loss. This means that a fiber with smaller core (such as G#2) can be expected to be more resistant against bend loss than a fiber with larger core (such as G#1), especially at shorter wavelengths. The fact that G#2 has lower loss than G#1 at shorter wavelengths could therefore be due just to smaller bend loss, and it is clear that comparing fibers with significantly different core diameters, as is done in Fig. 3(b), is a very bad idea when one is trying to prove that there is a difference due to surface scattering loss. Other geometrical differences between G#1 and G#2 could also play a role, but the authors provide no information about, for example, the cladding tube diameters or the gap size between them. So the statement in line 277 (“The distinction between the loss behaviors of fiber in G#1 and G#2 is, therefore, the optical manifestation of the SSL reduction in G#2 fibers”) is clearly a claim without any scientific foundation at all.

In attention to the reviewer's concerns, we added new graphs in Fig. 4 (formerly Fig. 3) to explain that, even though the core diameters of the fibers are different, one can correlate the difference between the measured loss trends to the diminishment of the surface scattering loss and, hence, to the improvement of the quality of the core surfaces of the fibers fabricated by following our new methods. In Fig. 4b, we now show a plot in which we multiply the loss figures shown in Fig. 4a by R_{co}^4 , where R_{co} is the radius of the fiber core. Multiplication by such a factor allows normalizing the loss with respect to the core size, as studied in the scaling laws available in reference [31] in the manuscript text. We observe that the loss normalization shown in Fig. 4b further clarifies the difference between the loss trends of the fibers in G#1 and G#1.

We also clarify that we are aware of the impact of bend loss in our fibers' performances and, to minimize its influence, we endeavored to realize our measurements with a considerably large curvature radius ($R \sim 1m$). This information is now available in the "Materials and Methods" section. Additionally, in the new Fig. 4, we present simulations of the fiber CL (thus considering the two fiber geometries with their corresponding core sizes and cladding tubes gaps) with integrated bending loss for a bending radius of 1 m. As we referenced in the answer to the reviewer's question #4, the analyses based on assessing the different loss mechanisms in our fibers allowed attaining an estimation of the SSL reduction which is remarkably consistent with the results of the surface roughness measurements.

16) Figure 4(c): the dashed line showing the *solid-core* silica Rayleigh scattering limit (SRSL) is misleading to be included in this way when comparing with loss in *hollow-core* fibers. If including such a comparison of loss in hollow-core fibers with the SRSL, it would only be fair to multiply the SRSL with the field-glass overlap ($\ll 1\%$) to get closer to the actual loss that can be expected in a hollow-core fiber from scattering in the glass. It would be more correct to instead compare with the surface scattering loss (SSL) as shown for example in Fig. 3(a). Why did the authors not instead plot the SSL curve from Fig. 3(a) in Fig. 4(c)? That would be a much more logical and fair comparison. Interestingly, when considering the theoretical total loss (TL) in Fig. 3(a), it is actually considerably lower than all experimentally achieved loss values here ("Fiber A" and Fiber "B"), which is actually a good indication that the scattering factor η (set to either 300 or 2100 in Fig. 3(a)) was not significantly lowered in Fiber A and Fiber B, despite the attempt to increase shear stress by applying vacuum to the core.

We clarify that the presentation of the silica Rayleigh scattering limit in Fig. 5c (formerly Fig. 4c) stands for an element of contextualization of our results within the fiber optics framework. Indeed, the silica Rayleigh scattering limit has been used as a benchmark to situate the achievements of the HCPCF community, as we can see in papers written by different research groups working on the demonstration of HCPCFs' performances (*e.g.*, the references [12], [14], and [16] in the manuscript text). Thus, we believe that it is worth keeping the silica Rayleigh scattering limit as a reference line for the contextualization of our results. In any case, in the new Fig. 4d and Fig. 4e, the reader can now find the SSL curves corresponding to the representative fibers in G#1 and G#2.

Additionally, regarding the reviewer's concern about the reduction of the surface scattering loss in our fibers, we refer to our answer to the reviewer's question #4, where we explain that we added new simulations on the different loss mechanisms in the fibers and confirm that fibers in G#2 have indeed lower SSL levels than fibers in G#1.

17) Another problem with the comparison in Fig. 4(c) between earlier work and "Fiber A" and "Fiber B": how can the authors be sure that none of the fibers presented in the earlier work,

were also not drawn while applying vacuum to the core? I could easily imagine that some of the earlier work also applied vacuum to the core, but that the fibers had higher loss for other reasons (for example, different core diameter, bend loss, etc.).

Indeed, we are not aware of all procedures that other groups might have used in earlier works. Thus, we designed our analyses and verified the effectiveness of our approach by comparing two sets of fibers fabricated by us. Fig. 5c (formerly Fig. 4c), on the other hand, aims to contextualize our results within the framework of HCPCF state-of-the-art (with no mention of the fabrication procedures used by other groups). The purpose of Fig. 5c is, therefore, to highlight our results as new record low loss values within the HCPCF technology scenario regardless of the fiber design, geometrical parameters, and, perhaps, different fabrication procedures.

18) Line 334-335: "In the setup for loss measurements in the UV range, one placed a flip mirror in-between the fiber end and the spectrometer so to allow correct positioning of the fiber output by observing its image in a CCD camera." This description is somewhat confusing, don't you rather mean that you observed the output on the CCD camera to ensure that you maximized coupling into the core at the *input* end of the fiber? The "correct positioning of the fiber output" is confusing without more explanation.

To account for the transmission spectrum of the fibers at wavelengths smaller than 400 nm, we have directed the output beam of the fiber to a spectrometer. As the spectrometer measurements' results are dependent on the alignment of the beam directed to it, a flip mirror has been placed in between the fiber output and the spectrometer window to assure the correct placement of the fiber output with respect to the latter. While the beam reflected by the flip mirror has been projected onto a CCD camera (which worked as a target for beam alignment), it allowed for correcting the fiber output position after the cutting procedures during the loss measurements. Thus, the mention in the text is correct, we did use the above-mentioned configuration to assure adequate positioning of the output fiber beam with respect to the spectrometer window. In attention to the reviewer's comments, we have included clarifications on the procedure described above in the "Materials and Methods" section in the new version of the paper.

In conclusion, the fibers presented here have interestingly low loss in the UV-VIS range, but there is no strong evidence for what the authors are trying to claim: that applying vacuum to the core during fiber drawing will lead to lower surface roughness and thereby lower loss. The comparisons between fibers cannot be made using fibers with significantly different geometries, because the impact of the geometry on confinement loss and bend loss, not to mention even on the surface roughness through the perimeter W , cannot be ignored. Even the drawing conditions (furnace temperature, drawing speed, etc.) will change T_G and thus the surface roughness.

We thank the reviewer for the professional assessment of our work and the relevant comments. We hope that our clarifications provided above adequately addressed the reviewer's concerns and appropriately informed the diligence and rigor that our work has encompassed.

REVIEWER COMMENTS

Reviewer #1 (Remarks to the Author):

In the revised manuscript, the authors addressed some of the issues previously raised, but the following issues remain:

1. The authors provide a new simple model to illustrate the effect of longitudinal gas pressure gradient and size variation on shear rate. However, the model does not capture the effect of counter-directional gas flows on surface roughness during fiber fabrication, which reflects the effect of transverse air pressure variations in different regions of the preform. So, the paper still lacks a deeper analysis of how the counter-directional gas flows can reduce scattering.

2. From Figure 4(b), the authors multiplied a factor to provide a more clearly the difference between the loss trends for comparison fibers. However, the citation [31] indicates that this coefficient is used to fit the effect of CL on the loss of the fibers. It is inaccurate to use this coefficient to fit the total loss in the normalised case of the fiber structure. Meanwhile, the simulation results in Figure 4(d) and Figure 4(e) show that microbending losses still dominate in the spectral region less than 600 nm. Other attenuation mechanisms may shift the low loss band of fibers with smaller core sizes to shorter wavelengths, regardless of scattering losses. Therefore, The statements in lines 351-352 are still not rigorous, and the authors should use fibers with similar structural parameters for comparison.

3. In Figs. 4(d) and 4(e), it is inaccurate to determine the scattering coefficients of the two fibers by matching the simulated TL with the measured loss, because similar total simulated losses can be obtained by simultaneously adjusting the scattering coefficients and the microbending fitting parameters of the fibers. Especially from the simulation results it can be out that at short wavelengths, the microbending loss plays an important role. It seems that the scattering coefficients of the two fibers are chosen just to corroborate the surface roughness measurements.

Reviewer #2 (Remarks to the Author):

The authors have satisfactorily made many very good changes to their manuscript. Although I don't agree 100% with everything (see 9 and 12 below), I think the manuscript can be published as is, although I would recommend that the authors still take my comments into consideration.

Ad 1) The toy model, text and figures added to the section "Shear stress as means to structure the surface roughness profile" are exactly what was needed to support the claims in the manuscript. Very well done.

Ad 2-7) Very good.

Ad 8) The authors correctly label one of the graphs in Fig. 1(d) with "coth", but in the text (line 161) there is a small typo saying instead "cosh". In addition, in line 163 the authors write "hyperbolic cosine term", where it should be "hyperbolic cotangent". Due to this confusion, I think it is imperative that the authors double-check whether they really used coth and not cosh for the calculation of the green dashed line in Fig. 1(d).

Ad 9) It is still not clear to me why the value of $T_G/\gamma = 2000 \text{ K} \times \text{m}^2/\text{J}$ (from $T_G = 1500 \text{ K}$ and $\gamma = 0.7 \text{ J/m}^2$) is considered correct for drawing the plot in Fig. 1(d), but not when

calculating the rms height of the frozen fluctuations in lines 171-172. If the authors believe that $\gamma = 0.7 \text{ J/m}^2$ is the most correct value (as determined from measurements on drawn silica capillaries in Ref. [2]) then this value must also be correct to use for the rms calculations in lines 171-172. The authors claim that "using the 0.4 nm rms roughness level to contextualize our results allows the reader to compare our results with an established benchmark in silica surface roughness studies". But it does not seem more important to be able to compare with some arbitrary benchmark for silica surface roughness studies in other fields, it seems relevant only to use the value of γ which is relevant for silica drawn under typical fiber drawing conditions. So the authors should stick to one value of γ (0.7 J/m^2) and use it for all calculations, as long as they are studying silica surfaces coming out of the fiber drawing process.

Ad 10-11) Very good.

Ad 12) I am not 100% sure if roughness at higher spatial frequencies can be neglected, just because the PSD figures are orders of magnitude lower; I think the spatial frequency relative to the wavelength of the light should also be important! For example, if I try to look through a window, it would be no problem for me if the window has huge deformations on a long length scale (small spatial frequencies) relative to the wavelength of the light. I can for example look easily through a large glass sphere (having a curvature much larger than the wavelength of light), although the light is refracted, and the glass sphere therefore may act as a lens. Nevertheless, the light is just redirected in a predictable way, and the light is not really lost. I could for example also use a mirror to redirect the light back through the glass sphere, and it would come out the same way as it went in, so the whole process is in a sense reversible.

On the other hand, if the window has surface roughness on the length scale of visible light, even if the amplitudes of this roughness would be small, the light would be scattered in all directions and lost. I would not really be able to place a mirror in front of the window to reversibly send the light back out through the window.

My analogy may not be 100% applicable here, but I just want to demonstrate that one needs to be very careful in assuming that roughness at high spatial frequency can be ignored just because the amplitudes (or equivalently, the PSD) of roughness at high spatial frequency are very small compared to the roughness at low spatial frequency.

I believe the analogy and concerns above are both captured by Eqn. (3) of Ref. [2].

Ad 13) Ok.

Ad 14) See my concern in 12).

Ad 15-18) Very good.

Answers to the reviewers

Reviewer #1 (Remarks to the Author):

In the revised manuscript, the authors addressed some of the issues previously raised, but the following issues remain:

We thank the reviewer for assessing our manuscript. In the following, we provide the answers to the reviewer's concerns. We hope that our clarifications adequately address the issues raised by the reviewer.

1. The authors provide a new simple model to illustrate the effect of longitudinal gas pressure gradient and size variation on shear rate. However, the model does not capture the effect of counter-directional gas flows on surface roughness during fiber fabrication, which reflects the effect of transverse air pressure variations in different regions of the preform. So, the paper still lacks a deeper analysis of how the counter-directional gas flows can reduce scattering.

The goal of our simple model is to demonstrate that, with a judicious choice of the drawing parameters along with the preform and fiber dimensions, one can achieve enhanced shear rate conditions on the fiber microstructure. While we apply counter-directional gas flows as a necessary condition to achieve the desired fiber microstructure geometry, we regret that our wording was not sufficiently clear to inform that the counter-directional flows involved in the core surface roughness problem are the flow of the gas inside the fiber core (upwards) and the flow of the glass due to the fiber drawing (downwards). In the revised manuscript, we have made amendments to the text so as to clarify this issue. The changes are highlighted in the marked version of the article.

Regarding the reviewer's comments on the transverse dynamics, we mention that, in practice, we set the drawing parameters in such a way that the transverse dynamics is stationary and, to first order, decoupled to the longitudinal dynamics, which is our main interest in the paper.

Thus, considering the above-mentioned aspects, we respectfully disagree with the reviewer and deem that our model does capture the conceptual elements of the problem we investigate in the manuscript, namely the establishment of a flow due to pressure gradients in the fiber and the corresponding shear rate increment on the fiber microstructure. Indeed, we consider that the merit of our model is its simplicity in providing a picture of the physics involved in the process and a working range for the fiber drawing parameters without the need for more complicated rheology models, which would be too complex and could confuse the scope of the paper.

2. From Figure 4(b), the authors multiplied a factor to provide a more clearly the difference between the loss trends for comparison fibers. However, the citation [31] indicates that this coefficient is used to fit the effect of CL on the loss of the fibers. It is inaccurate to use this coefficient to fit the total loss in the normalised case of the fiber structure. Meanwhile, the simulation results in Figure 4(d) and Figure 4(e) show that microbending losses still dominate in the spectral region less than 600 nm. Other attenuation mechanisms may shift the low loss band of fibers with smaller core sizes to shorter wavelengths, regardless of scattering losses. Therefore, The statements in lines 351-352 are still not rigorous, and the authors should use fibers with similar structural parameters for comparison.

We clarify that the utilization of the CL scaling laws in Fig. 4b intends to normalize the dependence of the CL with respect to the core diameters of the fibers and allow the reader to readily identify the remarkable difference between the loss trends of the fibers in G#1 and G#2. Additionally, we clarify that this normalization has not been used in the fit of the total loss in Fig.

4b and Fig. 4e, as the reviewer seems to have considered. We provide further clarifications on the simulations of Fig. 4d and Fig. 4e in the response to the reviewer's question #3.

Moreover, in attention to the reviewer's comments, we rewrote the statement in lines 351-352. It now reads:

“The distinction between the loss behaviors of fibers in G#1 and G#2 shown in Fig. 4b is, therefore, a strong indicator of the SSL reduction in G#2 fibers, as one will further analyze in the following when considering the different loss mechanisms in HCPCFs.”

3. In Figs. 4(d) and 4(e), it is inaccurate to determine the scattering coefficients of the two fibers by matching the simulated TL with the measured loss, because similar total simulated losses can be obtained by simultaneously adjusting the scattering coefficients and the microbending fitting parameters of the fibers. Especially from the simulation results it can be out that at short wavelengths, the microbending loss plays an important role. It seems that the scattering coefficients of the two fibers are chosen just to corroborate the surface roughness measurements.

We clarify that, in our procedure for fitting the fiber total loss, the parameter η in the SSL expression is the only free parameter. The parameters in the MBL formula are fixed and determined by the fiber microstructure (propagation constants and coupling integrals) or set in accordance with previous literature ($C_0 = 1$, as in reference [32] in the manuscript text). The CL, in turn, has been calculated by considering the fibers' microstructures, as well established in the HCPCF community. We, therefore, clarify that the calculated contributions of the SSL to the total loss have not been chosen so as to corroborate our measurement results, but they stem from a fitting procedure performed over a wide spectral range that rigorously considers the different loss mechanisms in HCPCFs.

In any case, in attention to the reviewer's comment, we re-run our fitting routine using different values for C_0 in the MBL expression and accounted for the square root of the ratio between the η values for the fibers in G#1 and G#2 ($\sqrt{\frac{\eta_{G\#1}}{\eta_{G\#2}}}$), which is the metrics we used to establish a comparison with the surface roughness measurements. As one sees in Table 1, $\sqrt{\frac{\eta_{G\#1}}{\eta_{G\#2}}}$ is mildly affected when different C_0 values are considered. Therefore, we observe that considering different values of C_0 in the MBL expression minorly impacts our conclusions.

Table 1. $\sqrt{\frac{\eta_{G\#1}}{\eta_{G\#2}}}$ values obtained from the total loss fitting using different C_0 values in the MBL expression.

C_0	$\sqrt{\frac{\eta_{G\#1}}{\eta_{G\#2}}}$
2.86	2.7
1.00	2.7
0.50	2.7
0.10	2.3
0.00	2.3

We added the following comment in the manuscript text in attention to the reviewer's comments:

“Also, we mention that different C_0 values in the MBL formula have been considered (from 0 to 2.86 [33]), and the ratio of the rms roughness suppression was found to be in the range of 2.3 to 2.7.”

Reviewer #2 (Remarks to the Author):

The authors have satisfactorily made many very good changes to their manuscript. Although I don't agree 100% with everything (see 9 and 12 below), I think the manuscript can be published as is, although I would recommend that the authors still take my comments into consideration.

We thank the reviewer for assessing our manuscript and for the positive appreciation of it.

Ad 1) The toy model, text and figures added to the section "Shear stress as means to structure the surface roughness profile" are exactly what was needed to support the claims in the manuscript. Very well done.

Ad 2-7) Very good.

We thank the reviewer for the positive judgment on the new version of the manuscript.

Ad 8) The authors correctly label one of the graphs in Fig. 1(d) with "coth", but in the text (line 161) there is a small typo saying instead "cosh". In addition, in line 163 the authors write "hyperbolic cosine term", where it should be "hyperbolic cotangent". Due to this confusion, I think it is imperative that the authors double-check whether they really used coth and not cosh for the calculation of the green dashed line in Fig. 1(d).

We thank the reviewer for detecting our mistake. The typos have been corrected in the new version of the manuscript.

Ad 9) It is still not clear to me why the value of $T_G/\gamma = 2000 \text{ K} \times \text{m}^2/\text{J}$ (from $T_G = 1500 \text{ K}$ and $\gamma = 0.7 \text{ J/m}^2$) is considered correct for drawing the plot in Fig. 1(d), but not when calculating the rms height of the frozen fluctuations in lines 171-172. If the authors believe that $\gamma = 0.7 \text{ J/m}^2$ is the most correct value (as determined from measurements on drawn silica capillaries in Ref. [2]) then this value must also be correct to use for the rms calculations in lines 171-172. The authors claim that "using the 0.4 nm rms roughness level to contextualize our results allows the reader to compare our results with an established benchmark in silica surface roughness studies". But it does not seem more important to be able to compare with some arbitrary benchmark for silica surface roughness studies in other fields, it seems relevant only to use the value of γ which is relevant for silica drawn under typical fiber drawing conditions. So the authors should stick to one value of γ (0.7 J/m^2) and use it for all calculations, as long as they are studying silica surfaces coming out of the fiber drawing process.

We understand the reviewer's point of view. In attention to it, we removed the horizontal line corresponding to the TESR (0.4 nm) in Fig. 3e.

Ad 10-11) Very good.

We thank the reviewer for the positive judgment on the new version of the manuscript.

Ad 12) I am not 100% sure if roughness at higher spatial frequencies can be neglected, just because the PSD figures are orders of magnitude lower; I think the spatial frequency relative to the wavelength of the light should also be important! For example, if I try to look through a window,

it would be no problem for me if the window has huge deformations on a long length scale (small spatial frequencies) relative to the wavelength of the light. I can for example look easily through a large glass sphere (having a curvature much larger than the wavelength of light), although the light is refracted, and the glass sphere therefore may act as a lens. Nevertheless, the light is just redirected in a predictable way, and the light is not really lost. I could for example also use a mirror to redirect the light back through the glass sphere, and it would come out the same way as it went in, so the whole process is in a sense reversible.

On the other hand, if the window has surface roughness on the length scale of visible light, even if the amplitudes of this roughness would be small, the light would be scattered in all directions and lost. I would not really be able to place a mirror in front of the window to reversibly send the light back out through the window.

My analogy may not be 100% applicable here, but I just want to demonstrate that one needs to be very careful in assuming that roughness at high spatial frequency can be ignored just because the amplitudes (or equivalently, the PSD) of roughness at high spatial frequency are very small compared to the roughness at low spatial frequency.

I believe the analogy and concerns above are both captured by Eqn. (3) of Ref. [2].

In our opinion, the analogy used by the reviewer does not fully fit the real situation in inhibited-coupling HCPCFs for two reasons:

- 1) In the glass sphere, the incidence is close to the normal and localized in a well-defined interface whereas, in the HC fibers, grazing incidence takes place since the effective index of the fundamental mode is just 1E-4 to 1E-5 below the air line. Additionally, the effects of the roughness are distributed along the entire fiber length;
- 2) In the glass sphere, the set of waves modeling the scattering are plane waves that form a continuum of solutions comprising all the angle values; in HC fibers, this set is composed of the fiber modes, and the power exchange among them is modeled by the coupling mode theory (CMT) as reported in Ref. [2].

Therefore, by following the CMT approach as in Ref. [2], we observe that the power exchange rate is driven by the Power Spectral Density, PSD, $S(k)$, computed at the spatial frequency $k = \frac{2\pi}{\lambda}\Delta n_{eff}$, where Δn_{eff} is the effective index difference between two modes exchanging power. In the case of core modes and hole modes, according to Marcatili's formula, Δn_{eff} scales with λ^2 and, hence, k scales with λ . This means that the shorter the wavelength, the lower the spatial frequencies involved in the power exchange due to the surface roughness. We understand that this aspect is corroborated by the experimental results presented in the manuscript, as fibers with lower PSD at low spatial frequencies exhibited lower loss than fibers showing PSD with increased values at lower frequencies (despite the slight difference in the PSD values for high spatial frequencies).

Ad 13) Ok.

We thank the reviewer for the positive judgment on the new version of the manuscript.

Ad 14) See my concern in 12).

The clarifications on comment 12) have been presented above. We hope they adequately address the reviewer's concern.

Ad 15-18) Very good.

We thank the reviewer for the positive judgment on the new version of the manuscript.

REVIEWERS' COMMENTS

Reviewer #1 (Remarks to the Author):

All the questions have been answered and I think the manuscript can be accepted.

Reviewer #2 (Remarks to the Author):

I am happy with the replies and modifications made by the authors, and recommend publication.

Answer to the reviewers

Reviewer #1 (Remarks to the Author):

All the questions have been answered and I think the manuscript can be accepted.

Reviewer #2 (Remarks to the Author):

I am happy with the replies and modifications made by the authors, and recommend publication.

We thank the reviewers for assessing our manuscript and for the acceptance recommendation.